# A novel checkpoint pathway controls actomyosin ring constriction trigger in fission yeast

**Tomás Edreira, Rubén Celador, Elvira Manjón, Yolanda Sánchez***

Instituto de Biología Funcional y Genómica, CSIC/Universidad de Salamanca and Departamento de Microbiología y Genética, Universidad de Salamanca, Salamanca, Spain

**Abstract** In fission yeast, the septation initiation network (SIN) ensures temporal coordination between actomyosin ring (CAR) constriction with membrane ingression and septum synthesis. However, questions remain about CAR regulation under stress conditions. We show that Rgf1p (Rho1p GEF), participates in a delay of cytokinesis under cell wall stress (blankophor, BP). BP did not interfere with CAR assembly or the rate of CAR constriction, but did delay the onset of constriction in the wild type cells but not in the *rgf1Δ* cells. This delay was also abolished in the absence of Pmk1p, the MAPK of the cell integrity pathway (CIP), leading to premature abscission and a multi-septated phenotype. Moreover, cytokinesis delay correlates with maintained SIN signaling and depends on the SIN to be achieved. Thus, we propose that the CIP participates in a checkpoint, capable of triggering a CAR constriction delay through the SIN pathway to ensure that cytokinesis terminates successfully.

## Introduction

Cytokinesis in fission yeast, as in most eukaryotes, involves the function of a contractile ring of actin filaments and myosin-II (AR) placed between the poles of the mitotic spindle during cell division. CAR constriction promotes division furrow ingression, so that sister chromatids are segregated to opposing sides of the cleavage plane (*Cheffings et al., 2016*; *D'Avino et al., 2015*; *Glotzer, 2017*; *Meitinger and Palani, 2016*; *Pollard and Wu, 2010*; *Rincon and Paoletti, 2016*; *Willet et al., 2015*). The failure of this critical step may result in genome instability and contributes to tumorigenesis (*Ganem et al., 2007*; *Nähse et al., 2017*; *Normand and King, 2010*).

The CAR of *S. pombe* is composed of short actin filaments bundled and oriented largely parallel to the PM (*Kamasaki et al., 2007*), and contains myosin IIs that pull on the actin filaments (*Laplante et al., 2015*; *Mishra et al., 2013*). CAR dynamics is tightly regulated in space and time and can be divided into several steps including positioning, assembly, maintenance, constriction, and disassembly. Ring position is determined during interphase by a broad band of cortical cytokinetic precursor nodes localized around the equator (*Akamatsu et al., 2014*; *Moseley et al., 2009*). These nodes mature further at mitosis entry when the anillin Mid1p becomes activated by the polo kinase Plo1p (*Almonacid et al., 2011*; *Bähler et al., 1998*), which initiates the recruitment of additional cytokinetic factors, including the IQGAP Rng2, Myosin II heavy and light chains (Myo2p, Cdc4p, and Rlc1p), the F-BAR protein Cdc15p and the formin Cdc12p (*Laporte et al., 2011*; *Padmanabhan et al., 2011*). Upon mitotic entry, more Mid1p binds to the PM, anchoring ring proteins (and then the ring itself) to this structure (*Celton-Morizur et al., 2004*; *Sohrmann et al., 1996*).

Ring assembly takes place between 10 and 15 min after SPB separation driven by interactions between actin filaments assembled by Cdc12p and Myo2p in adjacent nodes (*Vavylonis et al.,*

\*For correspondence:
ysm@usal.es

**Competing interests:** The authors declare that no competing interests exist.

2008; *Wu et al., 2006*). Once formed, the ring is maintained until the completion of anaphase in an interval known as maturation that lasts ~10 min until the onset of ring constriction. During maturation, the CAR does not change in size or shape, but many proteins are exchanged with others from the cytoplasmic pool (*Pelham and Chang, 2002*; *Roberts-Galbraith et al., 2009*). Mid1p disappears, and new F-BAR proteins and partners (Imp2p, Pxl1p, Fic1p) (*Roberts-Galbraith et al., 2009*), together with unconventional myosin-II (Myp2p) (*Laplante et al., 2015*) and the glucan synthase Bgs1p (*Goss et al., 2014*) join the ring.

Ring maintenance, constriction and septum formation depends on the septation initiation network (SIN). This network localizes to the SPB throughout the cell cycle, but also accumulates at the division site immediately prior to cytokinesis and is thought to transmit the signal to the CAR (*Goyal et al., 2011*; *Johnson et al., 2012*; *Simanis, 2015*). The SIN signal begins with the activity of the GTPase Spg1p and involves a regulatory GAP complex, several scaffolds and a linear cascade of three kinases (Cdc7p, Sid1p, and Sid2p), in order of their activation. Substrates of the Sid2p kinase are the phosphatase Clp1p/Flp1p (*Mishra et al., 2005*), the formin Cdc12p (*Bohnert and Gould, 2012*), the SIN scaffold Cdc11p (*Feoktistova et al., 2012*), the kinesin Klp2p (*Mana-Capelli et al., 2012*), the SAD kinase Cdr2p (*Rincon et al., 2017*), the anillin Mid1p (*Willet et al., 2019*) and several others (*Grallert et al., 2012*; *Gupta et al., 2013*). Overexpression of Rho GTPase Rho1p, its GEFs Rgf1p and Rgf3p, as well as inactivation of Rho1p GAPs, partially rescues the lethality of *sid2* mutants at a low-restrictive temperature (*Alcaide-Gavilán et al., 2014*; *Jin et al., 2006*). Based on these results, it has been proposed that the SIN activates Rho1p, which in turn activates the Bgs enzymes (*Jin et al., 2006*). However, the SIN target(s) involved in septum assembly remain unknown nowadays.

Both the SIN activity and the CAR are required for septum biosynthesis and the septum in turn contributes to CAR stabilization and constriction (*García Cortés et al., 2016*; *Proctor et al., 2012*; *Thiyagarajan et al., 2015*; *Willet et al., 2015*; *Zhou et al., 2015*). As the CAR constricts at a constant rate, the PM ingresses from the cell surface at the medial division site, with the extracellular cell wall just outside the PM, and the contractile ring attached to the membrane on the cytoplasmic side. Thus, septum biosynthesis is coupled to CAR constriction in fission yeast (*García Cortés et al., 2016*; *Roncero and Sánchez, 2010*).

Rho GTPases are key regulators of the actin cytoskeleton and their role in cytokinesis has been well established (*Glotzer, 2017*; *Yoshida et al., 2006*). In addition, local activation of Rho GTPases at the cleavage furrow promotes contractile ring formation in *Dictyostelium* and metazoans (*Wagner and Glotzer, 2016*) and activates formins in *S. cerevisiae*, *Dictyostelium*, and metazoans (*Kühn and Geyer, 2014*). In *S. pombe*, they appear to have none of the previously described functions. Rho1p is involved in regulation of the septum cell wall, but it has not yet been determined whether Rho1p plays a role in the early steps of cytokinesis (*García Cortés et al., 2016*).

Fission yeast Rho1p is a functional homologue of human RhoAp and budding yeast Rho1p (*Nakano et al., 1997*). Rho1p-GTP binds and activates the protein kinases of the PKC family, Pck1p, and Pck2p (*Pérez and Rincón, 2010b*), which function upstream of the MAPK module (Mkh1p, Skh1p/Pek1p, and Pmk1p/Spm1p) of the cell integrity signaling pathway (CIP) (*García et al., 2009a*; *Ma et al., 2006*; *Sánchez-Mir et al., 2014a*; *Viana et al., 2013*). Rho1p is activated by three guanine nucleotide exchange factors (GEFs), Rgf1p, Rgf2p and Rgf3p, that catalyze the exchange from GDP to GTP, rendering the GTPase in an active state (*García et al., 2006a*; *Morrell-Falvey et al., 2005*; *Mutoh et al., 2005*; *Tajadura et al., 2004*). Rgf1p and Rgf2p localize to the septum area, however, to date there is no evidence of their involvement in cell division (*García et al., 2009b*; *G Cortés et al., 2018 García et al., 2006a*; *Morrell-Falvey et al., 2005*; *Mutoh et al., 2005*). Rgf3p-GFP forms a ring that follows CAR constriction (*Morrell-Falvey et al., 2005*; *Mutoh et al., 2005*; *Tajadura et al., 2004*) sandwiched by the membrane-bound scaffolds Mid1p, Cdc15p, and Imp2p on the outer side and by F-actin and motor proteins on the inner side (*McDonald et al., 2017*).

Among the Rho1p-GEFs, Rgf1p is the most prominent. Rgf1p is required for the actin reorganization necessary for cells to change from monopolar to bipolar growth during NETO (New End Take Off) and activates the β–(1, 3) glucan synthase (*García et al., 2006a*; *García et al., 2006b*). Rgf1p shuttles in and out of the nucleus in interphase, and is accumulated within the cell nucleus in response to replicative stress (*Muñoz et al., 2014*). Here, we investigate the role of Rgf1p and Rho1p in cytokinesis and show that stress affecting cell wall integrity prevents the completion of cytokinesis on time. This inhibition depends on a signaling pathway that involves the Rho1p GEF,

Rgf1p, the Rho-GTPase, Rho1p, the ScPkc1p orthologue Pck2p, and the MAP kinase Pmk1p, all components of the cell integrity pathway (CIP). Inactivation of this pathway in stressed cells causes defects in septation (*Sengar et al., 1997*; *Toda et al., 1996*; *Zaitsevskaya-Carter and Cooper, 1997*). Thus, we propose that the CIP signaling delays CAR constriction in response to cell wall perturbations to ensure that cytokinesis reaches completion only after the cell has adjusted to the new conditions.

## Results

### Rgf1p localizes to the contractile acto-myosin ring (CAR) and interacts with CAR components

As described previously, Rgf1p accumulates at actively growing tips during interphase and re-localizes to a central ring during mitosis (*García et al., 2006a*; *Morrell-Falvey et al., 2005*; *Mutoh et al., 2005*). To precisely determine the temporal regulation of Rgf1p localization in cytokinesis, we investigated Rgf1p localization in a strain co-expressing Rgf1p-GFP$^{Envy}$ (Rgf1p-Envy), Rlc1p-tdTomato (myosin regulatory light chain 1) (*Le Goff et al., 2000*) and Sad1p-DsRed (a component of the spindle pole body (SPB) tagged with DsRed). In fission yeast, the actomyosin ring is assembled before anaphase, whereas its constriction occurs after completion of anaphase (*Wu et al., 2003*). The medial Rgf1p-Envy signal detected by time-lapse microscopy appeared during anaphase and before the emergence of the primary septum (PS) stained with calcofluor (blankophor, BP) (*Figure 1A*, medium frame). During actomyosin ring constriction, Rgf1p-Envy was seen nearby the constricting CAR, as well as with the developing septum (*Figure 1A*). We analyzed by super-resolution microscopy the localization of Rgf1p and that of Rlc1p and compared them by densitometry of the images (*Figure 1B*). Rgf1p-Envy seemed to follow the Rlc1p-tdTom ring and localized with the advancing septum edge leaving behind a trail as division proceeded (*Figure 1B*). This means that the Rgf1-GFP ring was located outside of the contractile ring.

To determine whether Rgf1p medial localization was associated with the actomyosin ring, we examined its localization in a β-tubulin *nda3*-KM311 mutant that stalls mitosis with an assembled CAR at the non-permissive temperature (*Hiraoka et al., 1984*). Rgf1p-GFP was localized to the division site in these prometaphase-arrested cells (50% of the cell population arrested in prometaphase, n = 50; *Figure 1C*). Consistent with this result, Rgf1p-GFP localization to the division plane was severely compromised in mutants defective in CAR assembly, *cdc15-140* (a mutant in the PHC family protein Cdc15 [*Fankhauser et al., 1995*]), *cdc3-6* (a mutant in profilin), and in AR maintenance, *cdc11-119* and *sid2-250* (mutants of the SIN components Cdc11p and Sid2p, respectively), at the restrictive temperature (*Figure 1D*, *Figure 1—figure supplement 1*). Additionally, we generated double mutants lacking Rgf1p and the essential and non-essential components of the CAR, namely *cdc4-8* and *rlc1Δ* mutants of myosin II light chains, *cdc15-140* and *imp2Δ* mutants of PCH family members and *cdc12-112* mutant of the cytokinetic formin. The genetic interaction with myosin mutants was weak (*Figure 1—figure supplement 2*). Strikingly, we found that the conditions in which *cdc15-140* and *cdc12-112* were able to divide and form colonies at 30°C and 32°C, respectively, the double mutants lacking Rgf1p were inviable at these temperatures (*Figure 1—figure supplement 2*). Altogether, these experiments establish that Rgf1p starts to accumulate in fully formed contractile rings and before the detection of the septum ingression (Blankophor staining), and that it might collaborate with other cytokinetic proteins in proper cell division.

### Cytokinesis is temporally blocked in cell-wall-stressed wild-type cells but not in *rgf1Δ* cells

Next, we studied CAR constriction in wild type and *rgf1Δ* cells expressing Imp2p-GFP (CAR marker) and Sid4p-GFP (SPB marker) by time-lapse microscopy (*Figure 2A*). Time zero corresponds to the time of SPB separation (*Wu et al., 2003*). Wild type and *rgf1Δ* early anaphase cells contained normal acto-myosin rings. Moreover, cytokinesis in *rgf1Δ* measured from the onset of SPB separation until the end of CAR constriction, lasted on average 39.9 ± 3.4 min, a duration that is similar to that of the wild-type cells (mean value 41.5 ± 2.6 min) (*Figure 2A and C*). To visualize septum formation, cells were treated with blankophor (BP), which is a fluorescent dye that specifically binds to linear β−1,3-glucan at the primary septum (*Cortés et al., 2007*). Interestingly, the addition of BP onto

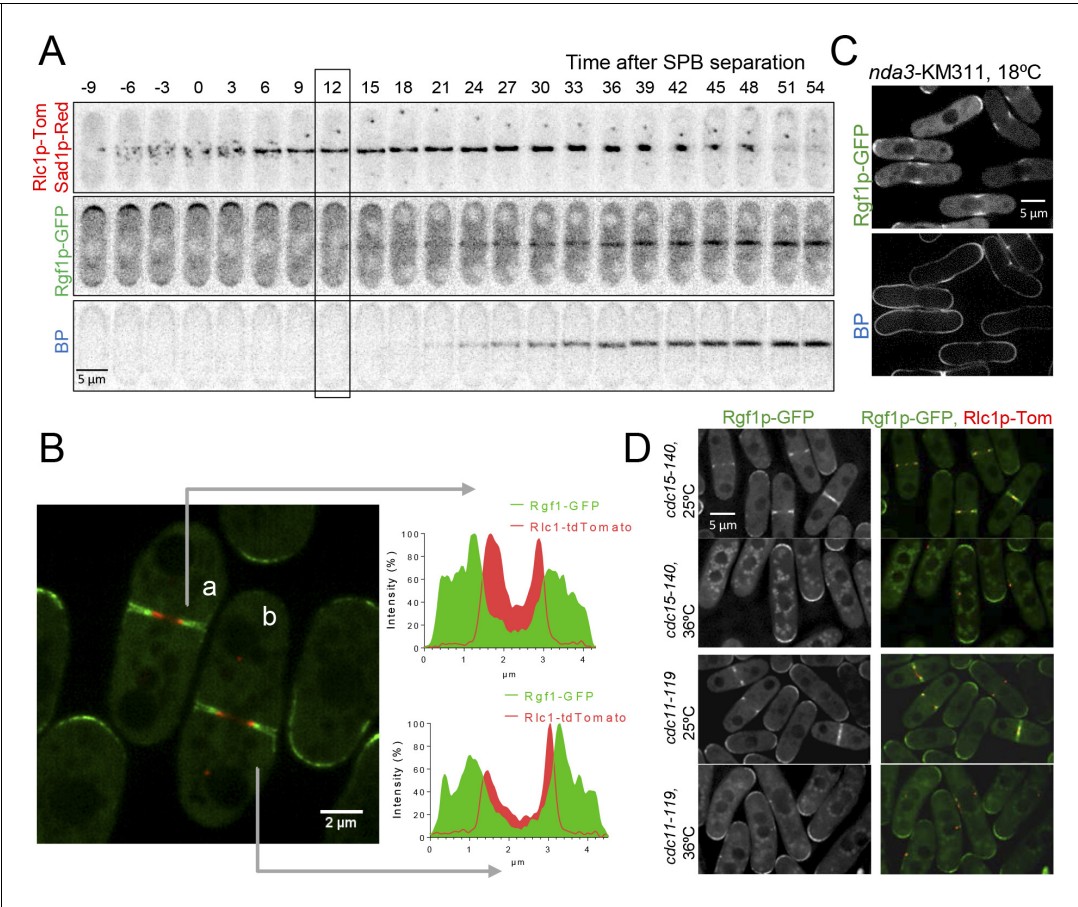

**Figure 1.** Rgf1p localizes to the contractile acto-myosin ring (AR) during cytokinesis. (A) Time series of fluorescence micrographs (maximum intensity projections of seven sections, inverted grayscale) of a single cell expressing three fluorescent fusion proteins: Sad1p-DsRed to label SPBs; Rlc1p-tdTom to label the ring; and Rgf1p-GFP. Blankophor (BP) 5 µg/ml was used to mark the septum. (B) Maximum projection of super-resolution images of the same cells as above showing the merged image of the localization of the SPBs (red), Rgf1p (green), and Rlc1p (red). The profile of green and red fluorescence intensity on a line across the division site is shown for cells a, and b, on the right panels. Scale bar 2 µm. (C) Cell images of *nda3*-KM311 strain expressing Rgf1p-GFP incubated at 18°C for 8 hr before imaging. Cells were stained with BP right before imaging. (D) Maximum projection images of *cdc15-140* and *cdc11-119* cells expressing Rgf1p-GFP and Rlc1p-tdTom. Cells grown at 25°C were transferred 3 hr at 36°C and the images were taken before and after the shift.

The online version of this article includes the following figure supplement(s) for figure 1:

**Figure supplement 1.** Rgf1p localization in CAR mutants.

**Figure supplement 2.** Genetic interactions between Rgf1 null mutants and CAR mutants.

wild-type cells caused a delay in the process of cytokinesis (mean value of 59.7 ± 6.7 min, *Video 1*). However, in the *rgf1Δ* cells cytokinesis lasted on average 48.1 ± 6.5 min, a duration which is more similar to the one recorded in the absence of BP treatment (*Figure 2B and C*, *Video 1*). Then, we performed a titration experiment to determine the time of cytokinesis in wild-type cells treated with different concentrations of BP. As shown in *Figure 2D*, BP caused a dose-effect type delay during our 2 hr movies at 28°C. Thus, for the rest of the experiments we used 5 µg/ml of BP to induce cytokinesis delay. To further test the effect of BP, we used another combination of markers, Rlc1p-tdTom and Atb2p-GFP (encoding α2-tubulin that marks microtubules) in *rgf1+* and *rgf1Δ* cells. As above, cytokinesis, measured from spindle formation until Rlc1p-tdTom signal closure at the cell equator, lasted longer in the wild-type cells than in *rgf1Δ* cells treated with BP (n = 15) (*Figure 2—figure supplement 1*). Under these conditions (the presence of BP), CAR contraction was completely blocked (rings did not initiate constriction at any time within the duration of the experiments) in 11 of the 61 wild-type cells examined. Conversely, none of the 83 *rgf1Δ* cells examined blocked cytokinesis

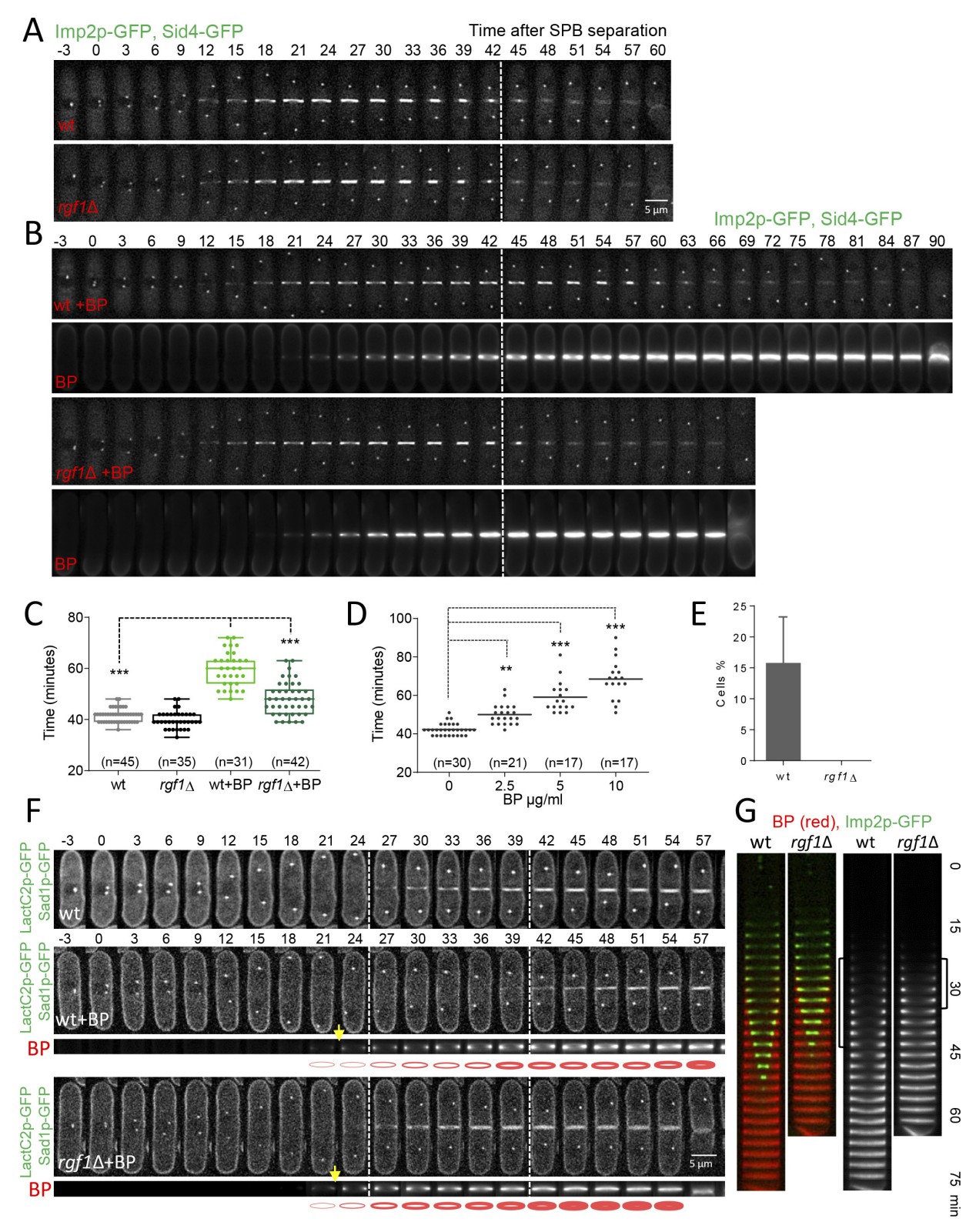

**Figure 2.** Blankophor (BP) prevents normal cytokinesis. (**A, B**) Timing of cytokinesis from SPB separation to the end of actomyosin ring contraction. Time-lapse series of wild type and *rgf1Δ* cells expressing Imp2p-GFP (AR marker) and Sid4p-GFP (mitotic marker). Cells grown at 28°C were imaged w/o Blankophor (**A**) or with Blankophor (5 µg/ml) added just before the time-lapse analysis (**B**). Images shown are maximum-intensity projections of z stacks. (**C**) Quantification of (**A and B**) is shown. Boxes represents the IQR, whiskers are the Tukey's range and the line inside the box is the median. Asterisks

*Figure 2 continued on next page*

*Figure 2 continued*

indicate the statistical significance of the difference between the two genotypes and the two conditions. Statistical significance was calculated by ANOVA followed by Šídák's multiple comparisons test (n.s. p>0.05; ***p<0.0001). (**D**) Blankophor (BP) slows down cytokinesis. Cells grown at 28°C were imaged during a time–lapse experiment in the presence of different concentrations of BP. Quantification of the timing of cytokinesis in individual cells of each BP condition is shown. Statistics were performed as in C, comparing each condition with untreated wild type cells (n.s. p>0.05; **p<0.005; ***p<0.0001). (**E**) The percentage of wild type and *rgf1Δ* cells treated with 5 µg/ml of BP, where the ring contraction is completely blocked. The data plotted here is the averaged of three independent experiments, with >20 cells for each strain, and the error bars represent the SD of the mean. (**F**) PM behavior during cytokinesis in BP treated cells. Wild type and *rgf1Δ* expressing Sad1p-GFP (SPB marker) and LactC2–GFP (PM marker) were treated with BP and time-lapse-imaged every 3 min. (t = 0 indicates SPB separation) (lower panels). BP fluorescence images are shown in the lower panels. Wild-type cells w/o BP are shown in the upper panel. (**G**) Kymographs showing the progression of Imp2-GFP and septum ingression as in (A) in wild type and *rgf1Δ* cells treated with BP.

The online version of this article includes the following source data and figure supplement(s) for figure 2:

**Source data 1.** Timing of cytokinesis in minutes from SPB separation to the end of actomyosin ring contraction.
**Figure supplement 1.** Timing of cytokinesis in wild type and *rgf1Δ* cells treated with BP.
**Figure supplement 1—source data 1.** Timing of cytokinesis in minutes from spindle formation until ring closure.
**Figure supplement 2.** Septum synthesis is not halted after BP treatment.
**Figure supplement 2—source data 1.** BP fluorescence intensity in the septum border over time, with time 0 at SPB separation.

(*Figure 2E*, *Video 2*). All together, these data indicate that BP alters the rate of cytokinesis and that this effect is less severe in cells lacking Rgf1p.

## Cell wall stress inhibits PM ingression coupled to CAR constriction

Cytokinesis requires association of the contractile ring with the PM and the cell wall, which in turn pushes the membrane for furrow ingression (*Proctor et al., 2012*). Defects in these linkages would result in a failure to draw in the PM and a concomitant delay in cytokinesis. In *S. pombe*, proteins like Cdc15p may contribute to the onset of this type of bridging function (*Roberts-Galbraith et al., 2010*). In addition, Cdc15p promotes the accumulation of glucan synthase (GS) Bgs1p (*Arasada and Pollard, 2014*), which also contributes to productive ingression. As mentioned above, BP binds to cell wall linear β–1,3-glucan and also interferes with its biosynthesis. Thus, we determined membrane progression and cell wall synthesis during BP treatment. Membrane ingression was followed in cells expressing the LactC2p–GFP fusion protein, which binds to phosphatidylserine at the PM (*Yeung et al., 2008*). In the absence of BP, the membrane began to ingress ~3 min after spindle breakdown (the moment when the two SPBs with maximum distance began to move toward one another). However, in cells treated with BP, ingression started ~18 min after spindle breakdown, a similar delay as the one detected in CAR constriction (*Figure 2F*, top panel). Furthermore, ingression was not delayed in *rgf1Δ* cells treated with BP (*Figure 2F*, lower panel). Thus, BP does not interfere with the linkage between the CAR and the membrane, but temporally inhibits membrane ingression coupled to CAR constriction in wild-type cells.

Interestingly, the BP signal concentrates at the cell equator forming a ring. This fluorescence became noticeable in anaphase, both, in wild type and in *rgf1Δ* cells, indicating that septum synthesis had started regardless of delayed CAR contraction *Figure 2F*, yellow arrows and (*G Cortés et al., 2018*). In the wild-type cells, the initial septum was maintained as a ring that did not progress toward the interior until further on. In *rgf1Δ* cells, the septum progressed more quickly causing the cells to separate earlier than the wild-type cells (*Figure 2F*). This event can be clearly seen in the kymographs of *Figure 2G*. From their analysis, it was also inferred that the blockage experienced by wild-type cells, did not halted septum synthesis. This is shown by the fact that BP signal intensity near the cortex

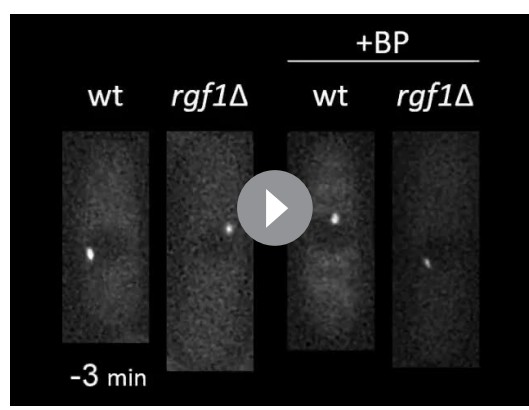

**Video 1.** Cytokinesis delay under BP treatment.
https://elifesciences.org/articles/59333#video1

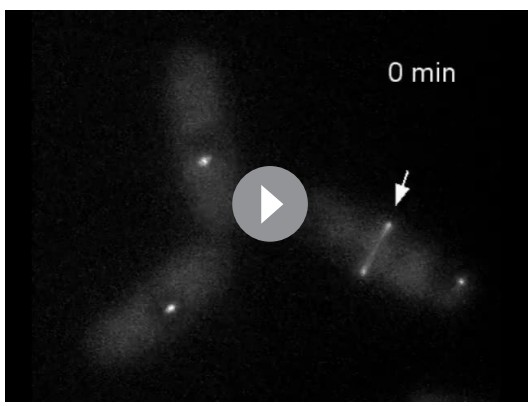

**Video 2.** Contractile ring blockage under BP treatment.
https://elifesciences.org/articles/59333#video2

increased over time until the onset of constriction (*Figure 2—figure supplement 2*). In sum, fluorescence microscopy showed that ingression of the ring (Imp2p-GFP), the PM and the stained septum are blocked during anaphase in wild-type cells treated with BP, and that this effect is much less apparent in the *rgf1Δ* cells. These results indicate that wild-type cells respond to cell wall stress with a delay in membrane ingression and septum progression and that Rgf1p is involved in this novel pathway.

## Rgf1p delays the onset of ring constriction in cell-wall-stressed cells

To pinpoint the role of Rgf1p in ring constriction, we followed the behavior of the contractile ring through the time-lapse analysis of *rgf1Δ* and *rgf1+* cells expressing mEGFP-Myo2p/Sid4p-RFP, in the presence or absence of BP (5 µg/ml). We analyzed three different stages of the contractile ring progression (*Figure 3A*): CAR assembly (time taken by the mEGFP-Myo2p nodes to coalesce into a ring with homogeneous fluorescence); CAR maturation (the interval between completion of contractile ring assembly and the onset of ring constriction); and CAR constriction (time between CAR diameter shrinkage and the complete closure of the ring). All statistical data are listed in *Table 1*. It was determined that the wild-type cells complete ring assembly within +18 min and +24 min, with a mean completion time of +21.1 ± 2.1 min (mean ± SD) in the absence of BP and 18.6 ± 2.9 in the presence of BP (*Figure 3A and B*). In the case of the *rgf1Δ* mutant, the average time required for the nodes to form a ring was similar to that of the one seen in the wild type either with or without (w/o) BP (*Figure 3B* and *Table 1*). Thus, Rgf1p does not contribute to contractile ring assembly, which is in accord with its late arrival, to the fully formed ring in wild-type cells (*Figure 1A*). In addition, BP does not alter the ring assembly time.

Next, we measured the amount of time that cells with fully assembled contractile rings spend before constricting. In the wild-type cells ring maturation lasted for approximately 10 min (10 ± 2.8 min, n = 16), while, in the same cells treated with BP the maturation was twice as long, lasting on average 22.1 ± 4.0 min (*Figure 3A and B*). Remarkably, the *rgf1Δ* cells did not exhibit such a prolonged maturation period in the presence of BP (untreated: 8.8 ± 2.5 min, n = 17; BP: 11.4 ± 2.4 min, n = 14) (*Figure 3A and B*). Finally, the time taken for the constricting rings to complete constriction was similar in both strains and conditions (*Figure 3A and B*).

To analyze the onset and the rate of constriction in more detail, first we monitored the time taken for Mid1p-mEGFP to be cleared from the CAR in *rgf1+* and *rgf1Δ* cells treated with BP. Mid1p–mEGFP dissociates from contractile rings during the maturation period (*Sohrmann et al., 1996*). According to the above results, Mid1p–mEGFP dissociated from contractile rings in 16.9 ± 3.1 min in *rgf1+*, by contrast the protein dissociated prematurely from the ring in *rgf1Δ* cells (12.0 ± 1.9 min) (*Figure 3—figure supplement 1*). Next, we followed contractile ring constriction in wild type and *rgf1Δ* individual cells marked with Imp2p-GFP/Sad1p-GFP in the presence of BP. In agreement with previous reports (*Ren et al., 2015*), Imp2p localizes to the CAR approximately 15 min after SPB separation, in both, wild type and *rgf1Δ* cells treated with BP (*Figures 2B* and *3C*). We measured the ring diameter over time in individual cells (*Figure 3C*), and these plots were used to quantify the onset and rates of ring constriction. The initial rings of Imp2p-GFP of wild type and *rgf1Δ* cells were similar in diameter (wild type: 3.56, *rgf1Δ*: 3.32 µm) and constricted at a similar rate,~0.15 µm/min. As pointed out above, the mean time for the onset of wild-type ring constriction after SPB separation was +37.9 ± 5.0 min, whereas rings lacking Rgf1p constricted at the cell-cycle time of +25.1 ± 3.6 min,~13 min earlier (*Figure 3C*, *Table 1*). Therefore, the presence of BP delayed the onset of constriction beyond the usual time set by the cell-cycle clock. However, the absence of Rgf1p allowed apparently normal ring constriction in BP stressed cells to take place. This behavior suggests that Rgf1p could be part of a mechanism that senses cell wall damage and delays ring constriction while activating repair processes.

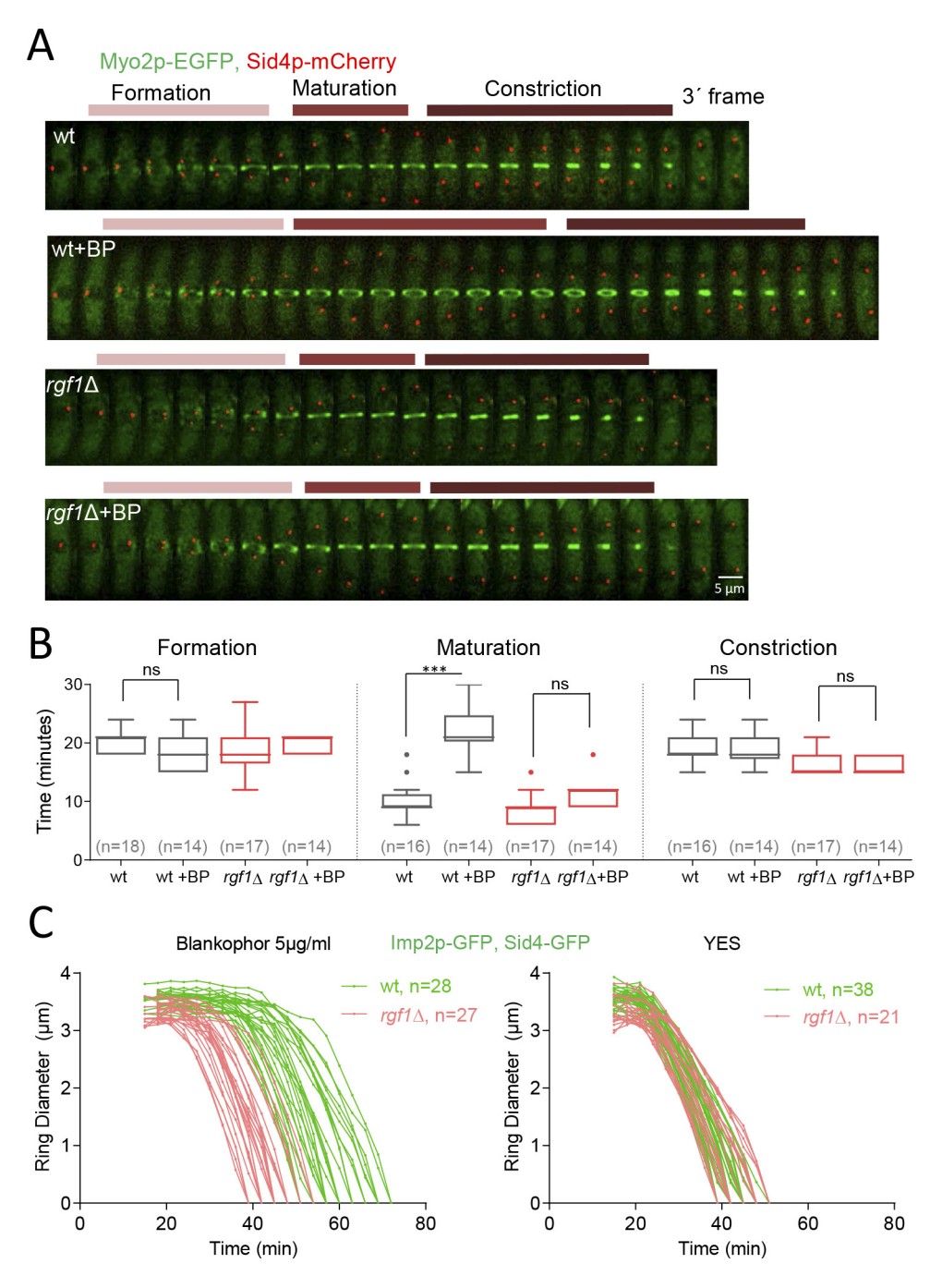

**Figure 3.** Rgf1p delays the onset of ring constriction in cell wall stressed cells. (**A**) Time lapse series of wild type and *rgf1Δ* cells expressing Myo2p-EGFP (AR marker) and Sid4p-mCherry (SPB marker). Cells were grown at 28°C and BP (5 µg/ml) was added before imaging, indicated in the panel. The images shown are maximum-intensity projections of z stacks. Scale bars represent 3 mm. (**B**) The time taken for various steps in cytokinesis (coalescence of nodes into a ring, dwell time before contraction, and contraction). Quantification of (**A**) was shown. Asterisks indicate the statistical significance of the difference between the two conditions. Statistical significance was calculated by ANOVA followed by Šídák's multiple comparisons test (n.s. p>0.05; ***p<0.0001). (**C**) Time course of the constriction of the CAR of wild type and *rgf1Δ* cells marked with Imp2p-GFP/Sad1p-GFP, in the presence (5 µg/ml) (right) or its absence (left) of BP, measured as the ring diameter over time, with time 0 min at SPB separation. The graphs show the diameters of each ring.

The online version of this article includes the following source data and figure supplement(s) for figure 3:

*Figure 3 continued on next page*

*Figure 3 continued*

**Source data 1.** Time in minutes taken for various steps in cytokinesis.

**Figure supplement 1.** Mid1p-EGFP dissociated prematurely from the ring in *rgf1Δ* cells.

**Figure supplement 1—source data 1.** Timing of Mid1-GFP disappearance from the ring .

## Rgf1p-dependent inhibition of abscission requires the cell integrity pathway (CIP)

Rgf1p signals upstream from the Pmk1p mitogen-activated protein kinase (MAPK) pathway (*García et al., 2009a*). This pathway regulates cell wall remodeling, cell separation, and ion homeostasis in injured cells (*Loewith et al., 2000*; *Madrid et al., 2006*; *Toda et al., 1996*; *Zaitsevskaya-Carter and Cooper, 1997*). Thus, we asked whether the CIP pathway was implicated in sensing BP. To this end, we measured the complete time of CAR constriction (from SPB separation until the end of constriction) in Pmk1p null mutants expressing Imp2p-GFP and Sid4p-GFP under BP stress conditions. While in wild-type cells, cytokinesis lasted a mean value of 59.7 ± 6.7 min in average, in *pmk1Δ* cells the mean value was 44.5 ± 2.8 min, mimicking the pattern of *rgf1Δ* cells (*Figure 4A and C*, *Video 3*). Moreover, like the *rgf1Δ*, *pmk1Δ* cells presented a shorter maturation time than that observed in wild-type cells in the presence of BP (*pmk1Δ* BP: 10.9 ± 3.1 min, n = 22; wt BP: 22.07 ± 4.0 min, n = 14) (*Figure 4—figure supplement 1* and *Table 1*). We also addressed whether this absence of response was due to a lack of kinase activity of Pmk1p under BP stress. *pmk1-KO* cells bearing a kinase-dead mutant version (Pmk1(K52E)GFP) (*Sánchez-Mir et al., 2012*), phenocopied the absence of cytokinetic response seen in the *pmk1Δ* in the presence of BP (*Figure 4—figure supplement 2*). These results indicate that Pmk1p signals the delay in the onset of constriction in cell-wall-stressed cells.

Besides the CIP, the stress-response MAP kinase pathway (SRP) affects cell separation under stress conditions. The SRP pathway promotes the transcription of specific gene cohorts and inhibits entry into mitosis in response to damage (*Chen et al., 2008a*; *Petersen and Hagan, 2005*; *Petersen and Nurse, 2007*). Additionally, the SRP has been reported to both promote and attenuate CIP signaling (*Loewith et al., 2000*; *Madrid et al., 2007*; *Madrid et al., 2006*). We checked whether Sty1p/Spc1p (the MAP kinase) was involved in BP sensing and obtained negative results. Cytokinesis in Sty1p null mutant cells lasted a mean value of 57.43 ± 5.1 min, which is similar than that of wild-type cells under BP stress conditions (59.12 ± 8.2) (*Figure 4—figure supplement 3*).

Next, we analyzed the functional significance of upstream components of the CIP pathway on the cytokinesis delay induced by cell wall stress (*Figure 4B*). The Rho GTPases, Rho1p and Rho2p, as well as some of its regulators (Rgf1p, Rga4p) and effectors (the protein kinase C PKC orthologues Pck1p and Pck2p) channel different stresses to the MAPK module. However, it is still not well understood which set of components participate in response to each of the initial triggering events. The Rho2p–Pck2p branch is fully responsible for MAPK activation in response to hyper- and hypo-osmotic stresses and both Rho1p and Rho2p target Pck2p for signaling cell wall damage to the CIP. However, the Pmk1p activation observed during cell separation or after treatment with hydrogen peroxide did not involve Rho2p–Pck2p (*Barba et al., 2008*; *Madrid et al., 2015*; *Sánchez-Mir et al., 2014b*). Among the strains assayed, *rgf1Δ* cells (previously shown) and *pck2Δ* were refractory to the

**Table 1.** Contractile ring parameters.

| Genotypes | Formation | Maturation time (min) | Constriction time (min) | Constriction rate (nm/min) |
|---|---|---|---|---|
| Wild-type | 21.1 ± 2.1 | 9.80 ± 3.2 | 18.9 ± 2.6 | 154 ± 21 |
| Wild-type + BP | 18.6 ± 2.9 | 22.1 ± 4.0 | 18.9 ± 2.7 | 158 ± 31 |
| *rgf1Δ* | 18.4 ± 3.3 | 8.08 ± 2.5 | 16.8 ± 2.1 | 148 ± 25 |
| *rgf1Δ* + BP | 20.1 ± 1.4 | 11.4 ± 2.4 | 16.1 ± 1.5 | 175 ± 30 |
| *pmk1Δ* | 17.4 ± 1.9 | 9.79 ± 2.1 | 16.1 ± 1.7 | 175 ± 16 |
| *pmk1Δ* + BP | 17.1 ± 2.8 | 10.9 ± 3.1 | 16.0 ± 1.9 | 187 ± 15 |

Mean ± SD.

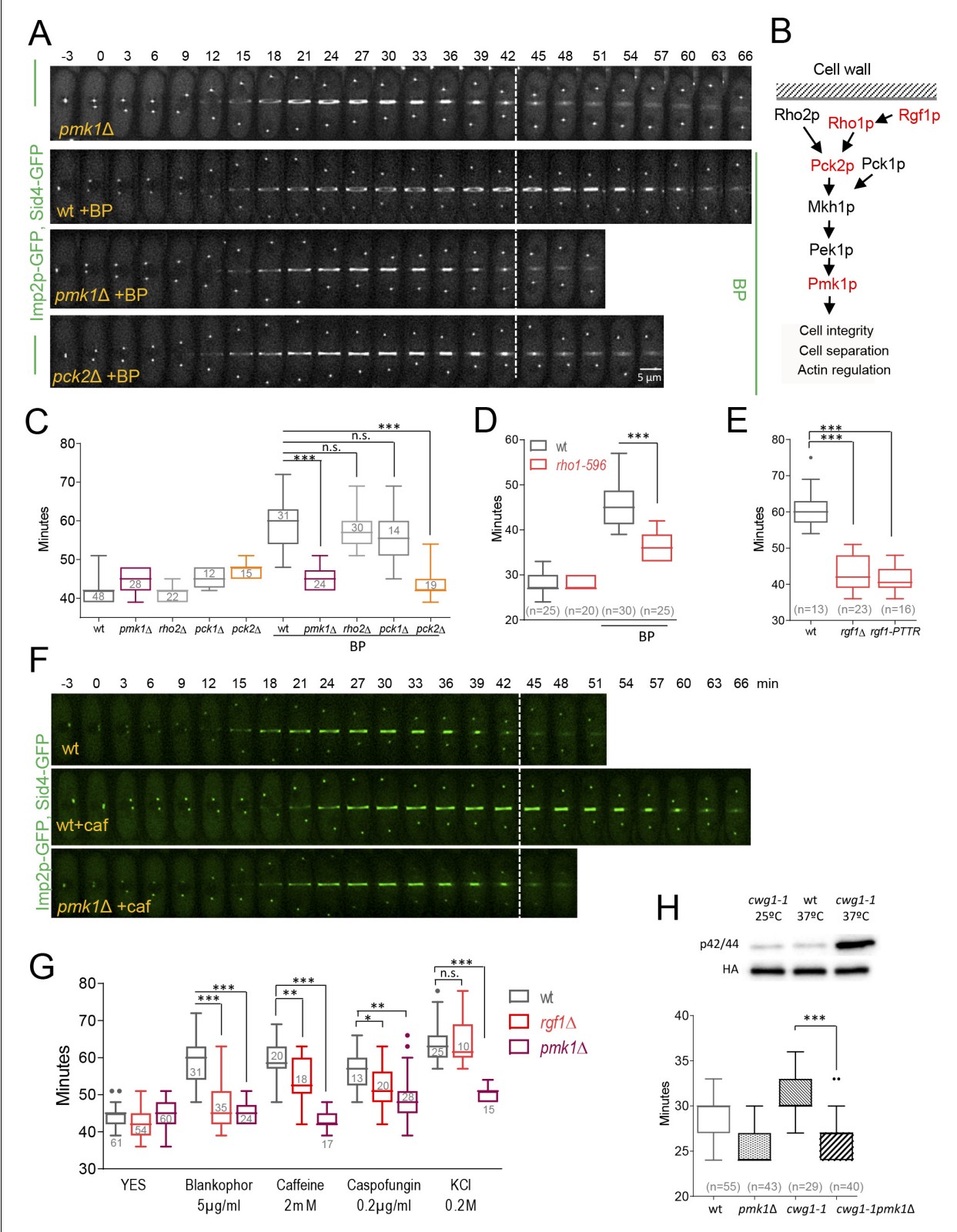

**Figure 4.** Cell wall stress-induced cytokinesis delay depends on Rho1p and on the cell integrity pathway (CIP). (**A**) A time-lapse series of wild type, *pmk1Δ* and *pck2Δ* cells expressing Imp2p-GFP (AR marker) and Sid4p-GFP (mitotic marker). Cells were grown at 28˚C, and BP (5 µg/ml) was added before imaging when indicated in the panel. The images shown are maximum-intensity projections of z stacks. (**B**) Scheme of the Pmk1p mitogen-activated protein kinase (MAPK) pathway (CIP). (**C**) Quantitation of the timing of cytokinesis (from SPB separation to the end of CAR contraction) in wild

*Figure 4 continued on next page*

*Figure 4 continued*

type, *pmk1Δ*, *rho2Δ*, *pck1Δ* and *pck2Δ* cells expressing Imp2p-GFP and Sid4p-GFP. Cells were grown at 28°C and a time-lapse analysis was carried out in the presence or the absence of BP (5 μg/ml) added before imaging. Asterisks indicate the statistical significance of the difference between the two conditions. Statistical significance was calculated by ANOVA followed by Šídák's multiple comparisons test (n.s. p>0.05; ***p<0.0001). (D) Quantitation of the timing of cytokinesis in wild type and *rho1-596* cells expressing Imp2p-GFP/Sid4p-GFP. Cells were grown at 25°C and shifted to 36°C 1 hr before time-lapse imaging at 36°C. BP (5 μg/ml) was added immediately before imaging. (t = 0 indicates SPB separation). Asterisks indicate the statistical significance of the difference between the two conditions. Statistical significance was calculated by Student's t test (***p<0.0001). (E) Quantitation of the timing of cytokinesis in wild type, *rgf1Δ*, and *rgf1-PTTR* cells expressing Imp2p-GFP/Sid4p-GFP. Cells were grown at 28°C and a time-lapsed analysis was carried out in the presence of BP (5 μg/ml), added before imaging. Asterisks indicate the statistical significance of the difference between the two conditions. Statistical significance was calculated by ANOVA followed by Fisher's LSD test (***p<0.0001). (F) Time-lapse series of wild type and *pmk1Δ* cells expressing Imp2p-GFP/Sid4p-GFP. Cells were grown at 28°C and caffeine (2 mM) was added before imaging when indicated. Images shown are maximum-intensity projections of z stacks. (G) Quantitation of the timing of cytokinesis in wild type, *rgf1Δ*, and *pmk1Δ* cells expressing Imp2p-GFP/ Sid4p-GFP as in C, in the presence or the absence of BP (5 μg/ml), caffeine (2 mM), caspofungin (0.2 μg/ml), and KCl (0.2M) added before imaging. Statistical significance was calculated by ANOVA followed by Fisher's LSD test (n.s. p>0.05; *p<0.05; **p<0.005; ***p<0.0001). (H) Upper panel, the wild-type strain and *cwg1-1* mutant, both carrying a *HA6H*-tagged chromosomal version of *pmk1+*, were grown in YES medium at 25°C and shifted to 37°C for 3 hr. Activated and total Pmk1p were detected with anti-phospho p44/42 and anti-HA antibodies, respectively. Lower panel, quantitation of the timing of cytokinesis in wild type, *pmk1Δ*, *cwg1-1* and *cwg1-1pmk1Δ* cells expressing Imp2p-GFP/Sid4p-GFP. Cells were grown at 25°C and shifted to 36°C 1.5 hr before time-lapse imaging at 36°C. Asterisks indicate the statistical significance of the difference between strains. Statistical significance was calculated by Student's t test (****p<0.0001).

The online version of this article includes the following source data and figure supplement(s) for figure 4:

**Source data 1.** Timing of cytokinesis in minutes from SPB separation to the end of actomyosin ring contraction.
**Figure supplement 1.** Pmk1p delays the onset of ring constriction in cells treated with BP.
**Figure supplement 1—source data 1.** Time in minutes taken for various steps in cytokinesis.
**Figure supplement 2.** Timing of cytokinesis in *pmk1-KO* cells treated with BP.
**Figure supplement 2—source data 1.** Timing of cytokinesis in minutes from SPB separation to the end of actomyosin ring contraction.
**Figure supplement 3.** Timing of cytokinesis in *sty1Δ* cells treated with BP.
**Figure supplement 3—source data 1.** Timing of cytokinesis in minutes from SPB separation to the end of actomyosin ring contraction.
**Figure supplement 4.** Timing of cytokinesis in *rgf1Δ pmk1Δ* cells treated with BP.
**Figure supplement 4—source data 1.** Timing of cytokinesis in minutes from SPB separation to the end of actomyosin ring contraction.
**Figure supplement 5.** Contractile ring constriction in wild-type cells treated with BP, caffeine, caspofungin, and KCl.
**Figure supplement 5—source data 1.** Ring diameter in μm over time, with time 0 min at SPB separation.

cytokinesis delay induced by BP, while *pck1Δ* and *rho2Δ* mutants behaved like wild-type cells (*Figure 4B and C*). Moreover, an *rgf1Δ pmk1Δ* double mutant exhibits no synergistic effects in the BP response (*Figure 4—figure supplement 4*), suggesting that the Rgf1p branch activating Pmk1p (*Figure 4B*) channels an efficient cytokinesis delay that protects cells from premature abscission under BP stress.

Because Rgf1p acts specifically on the Rho1p GTPase, we tested contractile ring dynamics in a hypomorphic and thermosensitive mutant of Rho1p, *rho1-596* (*Viana et al., 2013*), in the presence of BP at the restrictive temperature of 36°C. The role of Rho1p in the CIP is controversial and the available literature suggests that the function of Rho1p in the promotion of cell survival during cell wall stress involves both Pmk1p-dependent and -independent pathways (*Sánchez-Mir et al., 2014b*). Under BP treatment, cytokinesis proceeded faster in the hypoactive mutant (*rho1-596*) than in wild-type cells. However, no differences were found between these strains when left untreated (*Figure 4D*). Moreover, a deletion mutation in the RhoGEF domain of Rgf1p (*rgf1-PTTR*), which results in significantly reduced GEF activity towards Rho1p (*García et al., 2009a*), phenocopied the absence of ring constriction delay seen in the Rgf1p

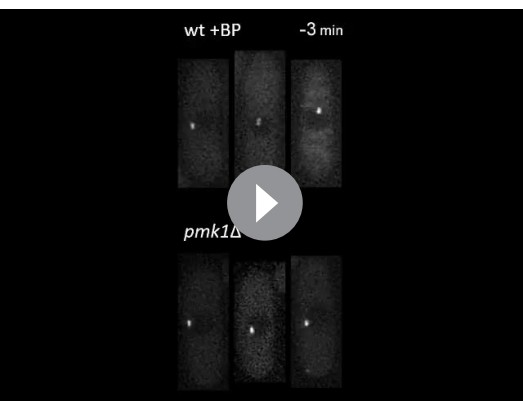

**Video 3.** *pmk1Δ* cells do not delay cytokinesis under BP treatment.
https://elifesciences.org/articles/59333#video3

deletion mutant in the presence of BP (*Figure 4E*). These results indicate that Rho1p is a key player in this process.

## Cell-wall-related stresses delay cytokinesis in a CIP-dependent manner

The aforementioned experiments demonstrate that BP blocks cytokinesis temporally in wild-type cells. Thus, we asked whether different treatments disrupting cell wall integrity or affecting the CIP also induce delayed cytokinesis. We used caspofungin (Csp), an inhibitor of the β−1, 3-glucan synthase (*Aguilar-Zapata et al., 2015*), and caffeine (Caf) that activates the Pmk1p cascade signaling in *S. pombe* (*Madrid et al., 2006*) and its orthologue in *S. cerevisiae* (*Kuranda et al., 2006*). Mild treatments with Csp (0.2 μg/ml) and Caf (2 mM) induced a cytokinesis delay in wild-type cells (*Figure 4F and G*, *Figure 4—figure supplement 5* and *Table 2*). In addition, we found that this delay was almost absent in *pmk1Δ* cells (*Figure 4F and G* and *Video 4*). In *rgf1Δ* cells, the observed delay was half way between that observed for *pmk1Δ* and wild-type cells (*Figure 4F and G*). The significance of the CIP in reacting to damage inflicted on cell wall structures was further supported by our analysis of the *cwg1-1* mutant cells, which hold a non-lethal thermosensitive mutation in the essential *bgs4*[+] gene (*Cortés et al., 2005*; *Muñoz et al., 2013*). We found that Pmk1p becomes phosphorylated in *cwg1-1* cells under restrictive conditions (3 hr at 37˚C) (*Figure 4H*). Moreover, this activation of the CIP correlates with a cytokinesis delay in the *cwg1-1* cells, which was also rescued by the elimination of Pmk1p (*Figure 4H*).

Finally, we addressed whether other adverse conditions that activate the MAPK Pmk1p, such as hyperosmotic stress, might also induce a cytokinesis delay. To this end, KCl (0.6M) was added to the medium to change the osmotic conditions during the time-lapse experiments. In the set-up, the cells shrank, the ring faded and growth temporarily ceased until turgor pressure had adapted (not shown) to the new environment. Under milder conditions, involving less KCl in the medium (0.2M), we did see a transient delay in cytokinesis (>15 min) (*Figure 4G* and *Figure 4—figure supplement 5*), which was independent on the presence of Rgf1p but dependent on Pmk1p (*Figure 4G*). Although this does not satisfactorily explain the reason for this difference, it could be possible that the other branch of the route, led by Rho2p, accounts for signal transduction under very mild changes in turgor pressure. On the other hand, *Proctor et al., 2012*, have shown that the differential osmotic pressure inside and outside the cell contribute to septum ingression. In these conditions, the phenotype inherent to the mutants (*pmk1Δ* and *rgf1Δ*) and the changes in the turgor pressure imposed by KCl, may explain the different observed behavior. Taken together, the data indicates that in response to mechanical stress, such as a deformed cell wall or stretching of the PM, the CIP temporally blocks CAR constriction.

## Mild cell wall stress conditions induce cell separation defects in *pmk1Δ* mutant

MAPK signaling pathways play a key role in eukaryotes to elicit proper adaptive responses to changes in the surrounding or adverse conditions (*Pérez and Cansado, 2010a*). Thus, we investigated the physiological consequences resulting from a compromised Pmk1p function in the presence of mild stress conditions (5 μg/ml BP, 0,6M KCl, 8 mM Caf); that is, similar conditions that induce an inhibition of CAR constriction in wild-type cells. In agreement with previous data (*Toda et al., 1996*; *Zaitsevskaya-Carter and Cooper, 1997*), the *pmk1Δ* cells were often shorter and rounder than wild-type cells and sometimes multiseptated cells (2%) were observed (*Figure 5A*). In the absence of stress, we observed a small increase in the septation percentage of *pmk1Δ* cells as compared to the wild-type cells (22% versus 17%) (*Figure 5A and B*). However, under BP treatment, the *pmk1Δ* mutant showed a substantial increase in the number of septated cells

**Table 2.** Contractile ring parameters in wild-type cells under stresses.

| | YES | BP(5 μg/ml) | Caffeine(2 mM) | Caspofungin (0.2 μg/ml) | KCl (0.2 M) |
|---|---|---|---|---|---|
| Mean Time Constriction Onset (min) | 21 ± 2 | 38 ± 5 | 27 ± 3 | 27 ± 3 | 27 ± 3 |
| Mean constriction Rate (nm/min) | 154 ± 21 | 158 ± 31 | 97 ± 17 | 114 ± 13 | 95 ± 17 |

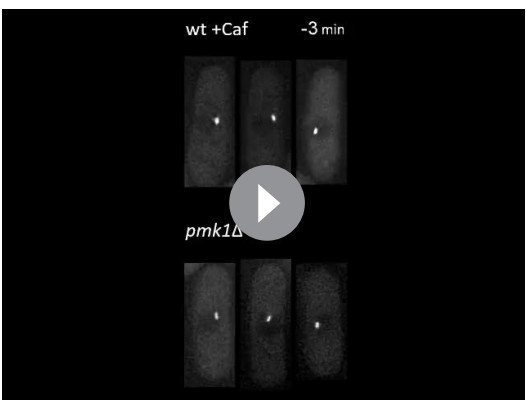

**Video 4.** *pmk1Δ* cells do not delay cytokinesis under Caf treatment.
https://elifesciences.org/articles/59333#video4

(42% with 1 septum plus 26% ≥ 2 septa, n = 386 compared to the number of septated cells in the *pmk1+* strain) under identical conditions (20% with 1 septum and 0 cells with ≥2 septa, n = 363) (*Figure 5A and B*). Moreover, a similar increase was detected after treatments with Caf and KCl (*Figure 5A and B*). As expected, multiseptated cells in the *pmk1Δ* mutant walled a nucleus per compartment, indicating that this phenotype was not the result of defective cell cycle control. Thus, the chained-cell phenotype was probably due to the mutant's inability to halt cytokinesis and induce the quality control machinery for septum assembly and/or disassembly.

## Effect of myosin and formin inactivation on the delay caused by cell wall stress

Ring constriction is achieved by molecular motors, actin cross-linking proteins and actin branching factors that generate constrictive forces that affect the actin filaments (*Laplante et al., 2015*; *Mishra et al., 2013*; *Palani et al., 2017*). To investigate whether this CIP-dependent cytokinesis delay was related to the regulation of the myosin II motor activity, we made use of an unconventional myosin-II Myp2p mutant that has a prolonged maturation period and a delay in the onset of ring constriction (*Laplante et al., 2015*; *Okada et al., 2019*; *Figure 6—figure supplement 1*). To do so, we treated *myp2Δ* mutant cells with 5 µg/ml of BP (to combine the effects of both methods that postpone ring contraction). Under this condition, the mean constriction time in the cells examined was considerably slower than in the control *myp2Δ* in the absence of BP (*myp2Δ* BP: 60 ± 5.9 min, n = 16, *myp2Δ*: 48 ± 5.5 min, n = 16). This suggested that the cell wall stress and Myp2 motor action act independently on the timing of cytokinesis. Similar results were obtained in cells lacking Rlc1p (Myosin light chain for Myo2p and Myp2p). These cells showed a considerable cytokinesis delay (*Sladewski et al., 2009*), which was aggravated in the presence of BP (*Figure 6—figure supplement 1*). These results suggest that these proteins do not bring on the ring constriction delay induced by cell wall stress.

In *S. cerevisiae,* the primary contractile force during budding yeast cytokinesis results from actin depolymerization mediated by cofilin and myosin II motor activity (*Mendes Pinto et al., 2012*). Thus, we explored whether the cytokinesis delay observed after cell wall stress was affected by mutations in the cofilin (Adf1p), required for actin severing (*Chen and Pollard, 2011*; *Nakano and Mabuchi, 2006*), and the formin Cdc12p involved in CAR polymerization (*Chang et al., 1997*). The *adf1-1* mutant exhibited a reduced rate in actin depolymerization in vivo (*Nakano and Mabuchi, 2006*). Then, we shifted *adf1-1* mutant at semi-restrictive temperature and monitored the time taken from SPB separation until completion of ring closure. Cytokinesis was slower in the *adf1-1* mutant compared to that of the wild type cells even without cell wall stress induction, indicating that the process was altered under these experimental conditions (*Figure 6—figure supplement 1*). Additionally, *adf1-1* cells showed a significant cytokinesis delay in the presence of BP as compared to their response to non-stress conditions (*Figure 6—figure supplement 1*). Similar experiments were performed using the *cdc12-112* mutant containing a point mutation in the FH2 domain that affects the rate of actin assembly (*Coffman et al., 2013*; *Figure 6—figure supplement 2*). We found that the *cdc12-112* cells shifted to a semi-restrictive temperature (34°C) and treated with BP act in response to cell wall stress (*Figure 6—figure supplement 2*). These results suggested that actin polymerization/depolymerization defects are not totally necessary for achieving a delay that is imposed by cell wall stress. However, the caveat exists that these temperature-sensitive mutant alleles at the semi-restrictive temperature may not represent a satisfactory loss of function. In fission yeast, Adf1p is more important for assembly than for CAR constriction (*Chen and Pollard, 2011*), while cell wall stress seems to affect the maturation steps, which may explain the additive results. Given that *cdc12-112* is affected in the catalytic domain (FHD1) (*Coffman et al., 2013*), the delay

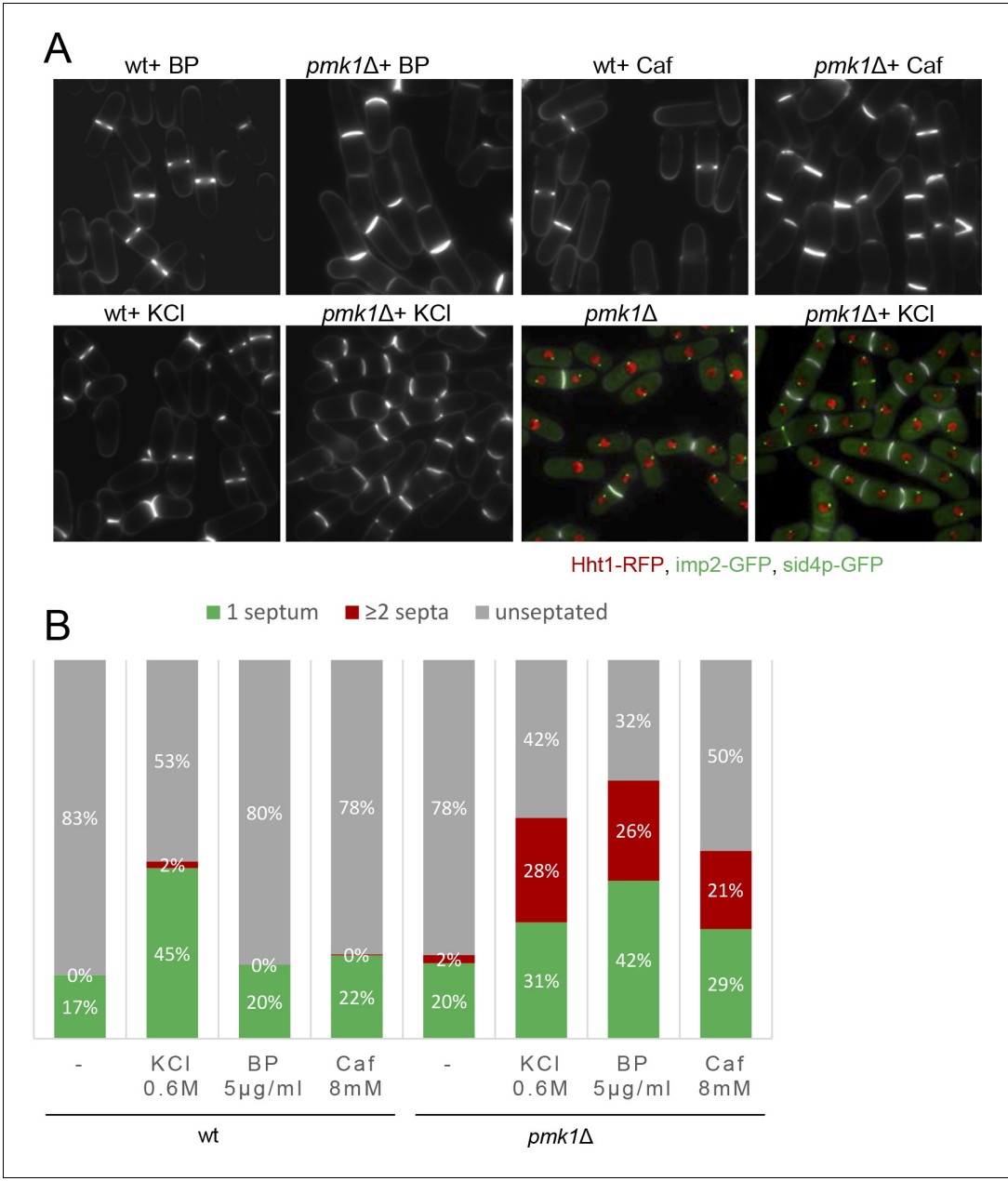

**Figure 5.** Mild cell wall stress conditions induce cell separation defects in *pmk1Δ* cells. (**A**) Asynchronous cultures of exponentially growing wild type and *pmk1Δ* cells were treated for 16 hr with KCl (0.6M), BP (5 µg/m) and caffeine (8 mM) at 31˚C and then stained with BP to visualize the septum. In the lower panels, the septa, the nucleus, the SPBs and the AR were visualized in wild type and *pmk1Δ* cells expressing Hht1p-RFP (histone H3, h3.1), Imp2-GFP (AR marker) and Sid4p-GFP (SPB marker) and stained with BP right before imaging, before or after KCl treatment. (**B**) Quantification of A is shown. The number of cells with one septum, ≥2 septa and unseptated cells was quantitated in live cells with or w/o treatment as indicated. Two hundred cells of each strain and condition were analyzed.

seen in the presence of BP could be additive to assembly or maturation defects inherent to the mutant.

## Effect of Clp1p inactivation on the delay caused by cell wall stress

In response to the perturbations of the actomyosin ring components, *S. pombe* cells arrest in a 'cytokinesis-competent' state, characterized by continuous repair and maintenance of the actomyosin ring (*Mishra et al., 2004*) and a G2 block. This checkpoint mechanism requires the function of the phosphatase Clp1p (a Sid2p substrate) (*Mishra et al., 2005*; *Trautmann and McCollum, 2005*) and the septation initiation network (SIN) (*Le Goff et al., 1999*; *Liu et al., 2000*). In response to cytokinesis defects, Clp1p, normally nucleolar in interphase, is retained in the cytoplasm until completion of cell division in a SIN-dependent manner (*Chen et al., 2008b*). Consequently, we asked whether BP treatments induce cytoplasmic retention of Clp1p. To this end, we performed a time-lapse analysis of cells expressing Clp1p-GFP, Imp2-GFP and Sid4p-GFP with and w/o cell wall stress (*Figure 6A*) and quantified the GFP nuclear signal intensity (*Figure 6B*). Clp1p was released from the nucleolus as cells entered mitosis. In agreement with previous reports (*Chen et al., 2008b*; *Mishra et al., 2005*), the protein was not fully re-accumulated in the nucleolus until some minutes after the CAR had finished constriction (~60 min after SPB separation). However, in cells perturbed with low doses of BP Clp1p was released from the nucleolus in a normal way but remained cytoplasmic during the resulting cytokinesis delay (*Figure 6A and B*). Thus, BP induces cytoplasmic retention of Clp1p. If Clp1p is involved in BP signaling, then, *clp1Δ* mutant cells treated with BP should not display the cytokinetic delay, as *rgf1Δ* and *pmk1Δ* cells do. However, the results shown in *Figure 6C* indicated just the opposite. In the cells examined, the mean constriction time was considerably slower than in the control *clp1Δ* w/o BP.

Clp1p is known to stabilize the acto-myosin ring in the presence of the actin depolymerizing drug Lat A (*Mishra et al., 2005*; *Trautmann and McCollum, 2005*). However, upon BP treatment, the rings appear to be more stable since they do not disassemble when they are blocked, even in the absence of Clp1 (*Figure 6C*). Thus, in this case, Clp1p would be unnecessary to stabilize the ring. To address the role of BP in the maintenance of the CAR more rigorously, we studied F-actin organization in wild type cells expressing Life Act–GFP endogenously (*Huang et al., 2012*). These cells were treated with a low dose of LatA (5 μM) in the presence and absence of BP (10 μg/ml) 30 min before starting the time-lapse analysis. In the event that BP could stabilize the ring, we predicted that the F-actin rings would be maintained for a long period of time in cells treated with Lat A in the presence of BP. By contrast, these rings would disappear more quickly in the untreated cells. The proportion of cells with medial F-actin structures was analyzed every 15 min in each condition. Interestingly, the wild-type cells treated with BP were able to maintain, for the duration of the experiment, a higher number of F-actin ring structures (2–3 folds) than the BP-untreated cells (*Figure 6D* and *Video 5*). This experiment establishes that upon mild perturbation of the cell wall during cell division the CAR becomes more stable.

In order to check the biochemical status of this stabilised ring, we analyzed the phosphorylation level of Cdc15p. Cdc15p becomes dephosphorylated within mitosis, peaking at anaphase, before it recovers its hyperphosphorylated form right after septation (*Clifford et al., 2008*; *Fankhauser et al., 1995*). Cdc15p dephosphorylation, partially dependent on Clp1p, induces its oligomerization, scaffolding activity and stabilizes Cdc15p localization to the division site (*Roberts-Galbraith et al., 2010*). We generated synchronous cultures by adding HU during 4 hr and released them in fresh medium at 25°C with or w/o BP. Cdc15p becomes dephosphorylated at the same time in the cells that carried BP than in mock cells. However, 120 min after release, we detected a reduced mobility shift of Cdc15p in the cells treated with BP compared to the mock cells (*Figure 6E* and *Figure 6—figure supplement 3*). Moreover, Pmk1p absence partially reverted the effect caused by BP in the electrophoretic mobility of Cdc15p, mimicking the kinetics of the untreated cells (*Figure 6E* and *Figure 6—figure supplement 3*). These results suggest that ring components are preserved in a stable state during the CAR constriction delay.

## Cell wall stress monitors CAR contraction through SIN signaling

Previous studies have shown that SIN mutants assemble actomyosin rings that are not maintained, and disassemble in late anaphase resulting in cytokinesis failure (*Simanis, 2015*). Thus, we

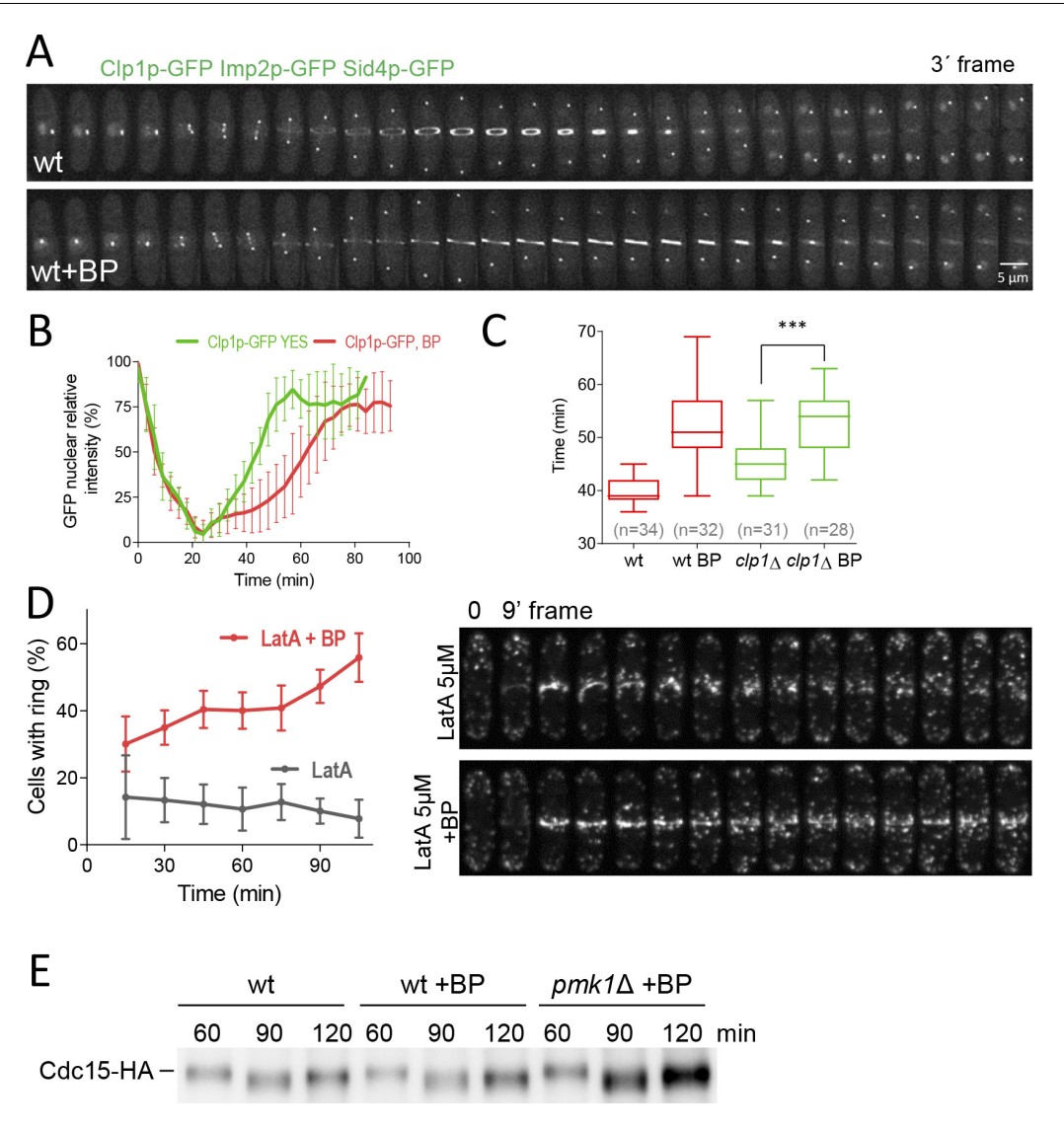

**Figure 6.** Dynamics of Clp1p-GFP and actin after cell wall stress. (**A**) Time-lapse images of Clp1-GFP cells expressing Imp2p-GFP and Sid4p-GFP were collected every 3 min using a DeltaVision microscope microscope. At each time point and position, seven slices were captured with a step size of 0.6 μm. Relative nuclear intensity of GFP over time, starting at SPB separation, is shown in B. (**C**) Quantitation of the timing of cytokinesis in wild type and *clp1Δ* cells expressing Imp2p-GFP/Sid4p-GFP. Cells were grown at 28°C and a time-lapse was carried out with or w/o BP (5 μg/ml), added before imaging. Statistical significance was calculated by ANOVA followed by Fisher's LSD test (***p<0.0001) (**D**) Wild-type cells expressing Life-act-GFP (*Huang et al., 2012*) grown to log phase at 28°C were treated with a low dose of LatA (5 μM) or with LatA (5 μM) and BP (10 μg/ml) and imaged during a time-lapse (left). The graph shows the percentage of cells with actomyosin rings. Note that in the cells treated with BP the LA-GFP rings are maintained for a prolonged period of time. (**E**) Protein extracts from Cdc15-HA synchronous cultures, treated or not with BP (1 mg/ml). Early-log phase cells were synchronized by adding HU (12.5 mM) for 4 hr, and then released to fresh medium with or w/o BP. At the indicated times after release, samples were analyzed by western blot using anti-HA antibodies.

The online version of this article includes the following source data and figure supplement(s) for figure 6:

**Source data 1.** Relative nuclear intensity of GFP over time, starting at SPB separation.

**Figure supplement 1.** Timing of cytokinesis in *myp2Δ*, *rlc1Δ*, and *adf1-1* mutant cells.

**Figure supplement 1—source data 1.** Timing of cytokinesis in minutes from SPB separation to the end of actomyosin ring contraction @31°C.

**Figure supplement 2.** Constriction initiation in *cdc12-112* cells.

*Figure 6 continued on next page*

*Figure 6 continued*

**Figure supplement 2—source data 1.** Time in minutes from SPB separation to the initiation of ring constriction @34°C.

**Figure supplement 3.** Cdc15p mobility shift in cells treated with BP.

considered the possibility that CAR stability in the presence of mild cell wall stress might be dependent on the SIN pathway. To address this question, *sid2-250* cells expressing Imp2p-GFP and Sid4p-GFP were shifted to a restrictive temperature and treated with BP (10 µg/ml) 30 min prior to being sampled at 2 and 3 hr. Interestingly, the number of rudimentary rings was similar in the presence or absence of BP (*Figure 7A*), suggesting that the SIN might be involved in CAR stabilization in the presence of cell wall stress.

It is known that localization of Sid2p to the division site, as well as the asymmetric SPB localization of Cdc7p to one SPB, are considered markers of maximal SIN activation and Cdk1p inhibition (*Dey and Pollard, 2018*; *Johnson et al., 2012*; *Wachowicz et al., 2015*). To visualize more closely the state of the SIN under the effect of cell wall perturbation, we imaged wild type (*Figure 7B*) and *rgf1Δ* cells in combination with these two SIN components (*Figure 7—figure supplement 1*). In a SIN 'early' state (*Wachowicz et al., 2015*), Cdc7p localized faintly to both SPBs and similar kinetics were observed in time-lapse experiments involving wild type cells with or w/o BP (*Figure 7B*). In the 'late' state, the localization of Cdc7p-GFP was asymmetric and lasted until the completion of cell division in both conditions (*Figure 7—figure supplement 1*). However, the peaked level of Cdc7p dropped after ~30 min in the untreated cells and was maintained at the new SPB for prolonged periods (up to 45 min) in the wild-type cells treated with BP (*Figure 7B* right and left panels). Sid2p was timely recruited to the ring in the wild type and *rgf1Δ* cells treated with BP, but the protein disappeared earlier from the division site in *rgf1Δ* than in the wild-type cells, just as ring constriction was finishing (*Figure 7—figure supplement 1*). Similarly, Cdc7p signal at the new SPB lasted longer in wild type than in *rgf1Δ* cells (*Figure 7—figure supplement 1*). These results strongly suggest: (1) that the delay in constriction induced by cell wall stress is not the consequence of low SIN activity, but instead correlates with a prolonged signal; and (2) that Rgf1p and the CIP may function to extend the duration of SIN signaling upon cell wall perturbations, thereby providing an improved opportunity to constrict the ring properly.

Following on, we asked whether the SIN pathway was required to achieve the cytokinesis delay induced after cell wall stress. At its semi-permissive temperature (32°C), the *sid2-250* strain, partially compromised in Sid2p function is capable of ring formation and constriction. This result, allowed ring dynamics analysis in the presence or absence of cell wall stress. Rings assembled normally in this mutant after 1 hr at 32°C (*Figure 7C*). However, these cells did not show the delay induced by cell wall stress (*Figure 7C*). Similar results were obtained using *sid1-239* at the semi-restrictive temperature (*Figure 7C*), suggesting that SIN signaling is involved in the control of CAR constriction in response to cell wall stress. Based on these studies, we conclude that the SIN is important for initiating the delay in CAR constriction imposed by cell wall stress.

## Discussion

The analysis of factors that trigger ring constriction is still one of the least developed aspects of cytokinesis (*Pollard, 2017*). Here, we show that cell wall damage inflicted during cytokinesis triggers a checkpoint-like response, promoting a delay right before CAR constriction. Sensors relay the signal to the small GTPase Rho1p, triggering phosphorylation events through the cell integrity pathway (CIP). Consequently, inactivation of pathway components by mutations leave the

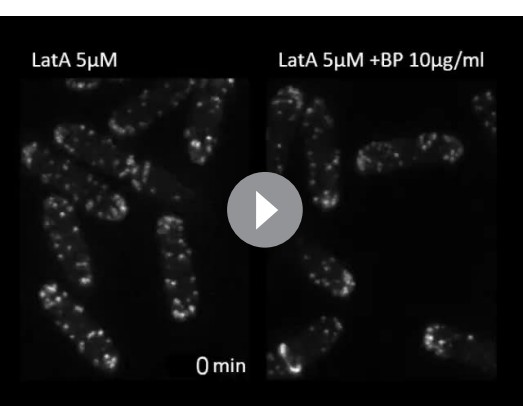

**Video 5.** Contractile rings become more stable under BP treatment.

https://elifesciences.org/articles/59333#video5

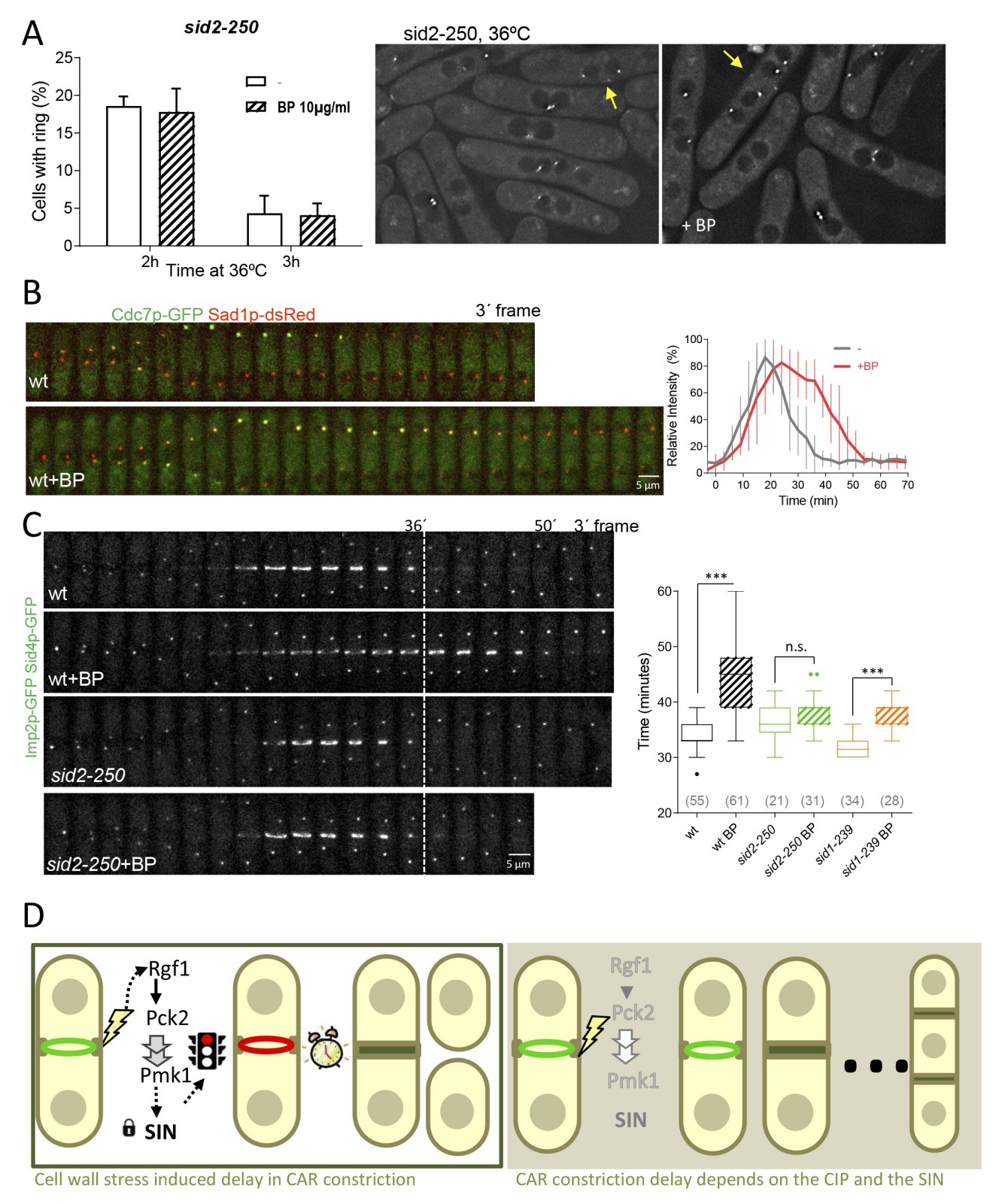

**Figure 7.** SIN signaling is required for cell wall stress dependent delay. (**A**) The graph shows quantitative data for *sid2*-250 cells expressing Imp2p-GFP/ Sid4p-GFP and grown to exponential phase at 25˚C. Cultures were shifted to 36˚C and imaged after 2 and 3 hr. Thirty minutes before imaging half of the cells were treated with BP (10 µg/ml) (right panels). (**B**) A time series of the maximum intensity projections of fluorescence micrographs of cells expressing (green) Cdc7p–EGFP and (red) Sad1p–tdTomato at 28˚C. Cells were imaged w/o BP (top panel) (n = 11) or with BP (5 µg/ml) added (lower

*Figure 7 continued on next page*

*Figure 7 continued*

panel) (n = 12) using a spinning-disk confocal microscope. Measurements of Cdc7p–EGFP intensity on the brighter SPB over time in cells with BP (red) or w/o BP (grey) shown in A, with time 0 being SPB separation. (C) Time lapse series of wild type and *sid2-250* cells expressing Imp2p-GFP and Sid4p-GFP. Cells were grown at 25°C and shifted to 32°C 1 hr before time-lapse imaging at 32°C. BP (5 µg/ml) was added immediately before imaging. (t = 0 indicates SPB separation). Quantitation of the timing of cytokinesis of strains and conditions shown in C. Asterisks indicate the statistical significance of the difference between the two conditions. Statistical significance was calculated by ANOVA followed by Šídák's multiple comparisons test (n.s. p>0.05; ***p<0.0001). (D) Model for CIP-mediated CAR contraction delay caused by cell wall stress in fission yeast cytokinesis. Typically, *rgf1*+ and *pmk1*+ cells undergo a cytokinesis arrest in the presence of mild cell wall stress. In this context, Pmk1p signal maintains CRs stability in a SIN dependent manner. However, when the signal is blocked in deletion mutants of the CIP components or when the SIN kinase Sid2p is not fully active, the ring is prematurely closed. Persistent cell wall stress leads to the formation of multiseptated cells. Thus, the CIP-mediated signal is transmitted to the SIN that stabilizes AR before constriction and enable proper CAR performance.

The online version of this article includes the following source data and figure supplement(s) for figure 7:

**Source data 1.** Percentage of cells with actomyosin rings over time, shifted to 36°C at time 0h.
**Figure supplement 1.** Cell wall stress delays Sid2p silencing in wild-type cells but not in *rgf1Δ* cells.
**Figure supplement 1—source data 1.** Time in minutes of the arrival and release of Sid2-GFP at the septum.

damage signal unchecked during cytokinesis, which results in premature CAR constriction and leads to the failure of cell cleavage (*Figure 7D*).

In this work, we addressed the question of whether Rgf1p and Rho1p play a role in cytokinesis in addition to their function in septum synthesis (*Pérez et al., 2016*). We confirmed and extended previous observations showing that Rgf1p localizes to the proximity of the assembled ring (*García et al., 2006a*; *Mutoh et al., 2005*); the protein follows the advancing septum edge leaving behind a trail as division proceeds (*Figure 1*). This localization differs from that of Rgf3p, which forms a ring that disappears at the end of CAR constriction (*Morrell-Falvey et al., 2005*; *Mutoh et al., 2005*), suggesting a role for Rgf1p in the furrow ingression step. Rgf1p localizes to the assembled ring before the septum is perceived. Interestingly, we found that Rgf1p controls the onset of ring constriction under cell wall stress (Blankophor 5 µg/ml), but does not affect CAR dynamics in unperturbed growth conditions. This is probably the reason why Rgf1p function in cytokinesis has been ignored for such a long time. Moreover, the requirement of Rgf1p to delay CAR constriction is due to the activation of Rho1p, as both the *rgf1-PTTR* and *rho1-596* mutants exhibited a lack of ability to delay cytokinesis similar to that of the *rgf1Δ* mutant.

Our work has also built upon what is already known about the contribution of cell wall synthesis to cytokinesis (*Proctor et al., 2012*; *Zhou et al., 2015*). A recent idea in the field of cytokinesis is that, for walled eukaryotes such as *S. pombe*, the dominant factor determining the rate of ring constriction is not the amount of tension that can be generated in the contractile ring, but rather the rate of synthesis of new cell wall material used to generate the invaginating furrow (*Cheffings et al., 2016*; *Proctor et al., 2012*). We have shown that treatments that disorganize the cell wall (BP) and delay septum progression also delay CAR constriction (*Figure 2*). This impact is considered significant, as 15% of the cells with assembled rings never go on to complete ring closure; the rest (~85%) show a delay of 12–15 min as compared to the untreated cells. Moreover, cell wall stress influences the onset of CAR constriction, but does not interfere with CAR assembly or the rate of CAR constriction. It is possible that in the temporary absence of a cell wall headway the CAR provides insufficient force to be closed. If BP reduces the force generated by the septum, then the cells could experience an imbalance in force production between the CAR and the septum and the CAR might need to provide additional force (*Proctor et al., 2012*). Alternatively, a regulatory pathway may couple constriction of the CAR and septum formation such that a defect in septum formation could prevent the constriction of the CAR. Some of these possibilities are not mutually exclusive. We have shown that cell wall stress (inflicted by BP) generates a delay of approximately 15 min in CAR constriction, and this has a direct impact on cell separation. Moreover, mutants in Rgf1p, Rho1p, Pck2p, upstream regulators, and Pmk1p, the MAP kinase of the cell integrity pathway, do not exhibit a delay in cytokinesis followed by mild cell wall stress. Other sorts of cell wall damage, such as that caused by a point mutant in the main β-GS, also lead to a delay in cytokinesis that is sensed by the CIP. In addition, mild treatments with antifungal agents like caspofungin and caffeine also behaved similarly.

Time is important. Although, it is known for a long time that the MAP kinase Pmk1p becomes activated 'within minutes' by cell wall stress (*Barba et al., 2008*; *Madrid et al., 2007*), it is still

unclear how the CIP integrates this stress input with successful cell separation (*Bähler, 2005*). A quick response is required when the cell's genome has already split and the cell becomes ready to separate its cytoplasm. Cell wall sensors/transducers detect the perturbation in the septum as it starts to form and activate Pmk1p to block cytokinesis progression under the situation of a deficient septum. The pathway can be equally impacted by modulating the phosphatase Pmp1p, which removes the activating phosphorylation on Pmk1p (*Sugiura et al., 1998*). In the case that the checkpoint is overridden, some septa could be poorly built up. In that sense it is known that conditions of nutrient limitation, hypertonic stress or caffeine, prompt *pmk1Δ* cells to grow as short-branched filaments in which the cell walls and septa are thickened (*Zaitsevskaya-Carter and Cooper, 1997*). At the same time, Pmk1p might induce the expression of genes that function to replace the damaged wall and restart cytokinesis even under stress conditions (*Chen et al., 2003*; *Takada et al., 2007*).

We considered how Pmk1p is able to accomplish CAR regulation after cell wall stress. The coordination mechanism may involve proteins that are physical components of the ring or regulatory proteins that may link the pathways together (*Cheffings et al., 2016*; *Pollard, 2017*). In this sense, we have shown that defects in formin and myosin II, that are force-generating components of the CAR, show a delay in CAR maturation (*Laplante et al., 2015*; *Lord et al., 2005*; *Stachowiak et al., 2014*), which was additive to the delay observed during cell wall stress. However, a mutation in the SIN kinase Sid2p was refractory to the delay caused by cell wall stress, indicating that the CIP may act by promoting the maintenance of SIN activity. There is a much evidence that reinforces this premise. First, cell wall stress correlates with a prolonged SIN signal, where Cdc7p remains on the new SPB for more time and the cytoplasmic retention time for Clp1p is longer (*Figures 6A* and *7B*). This does not occur in the *rgf1Δ* and *pmk1Δ* (not shown) mutants were the SIN signal is deactivated earlier (*Figure 7—figure supplement 1*); these mutants behave like wild-type cells in the absence of cell wall damage. Second, cell wall damage makes contractile rings more stable, where they become less vulnerable to mild perturbations of the actin cytoskeleton. Given that Sid2p is required for CAR maintenance when the cytokinesis checkpoint is active (*Alcaide-Gavilán et al., 2014*), it is very likely that the prolonged SIN activity serves to maintain the CAR in a competent state to achieve constriction safely. Third, SIN mutants grown at high temperature block the cell wall stress signal (*Figure 7A*). In these conditions, the rings are not maintained owing to a deficient SIN that is incapable of stabilizing them.

Cytokinetic furrow ingression requires both the actomyosin ring constriction and septum formation. Since one process cannot progress without the other, at least initially, cytokinesis can potentially be arrested through the blockage of either of these two components. There are two reasons that favor the proposal of a blocking model that operates through a halt in the onset of ring constriction. First, septum formation, measured as the increase of BP intake, is not stopped but increases linearly over time even without any trace of ring constriction (*Figure 2—figure supplement 2*); and second, the CAR becomes stabilized in a SIN-dependent manner after cell wall stress. In this study, we also address how this system is able to block cytokinesis when the ring is already formed, especially considering the onset of ring constriction is one of the least understood aspects of cytokinesis. Although ring constriction usually coincides with full SIN activation, we have shown a situation where the SIN is kept transiently active and the onset of constriction correlates with SIN inactivation. Similar to the cytokinesis checkpoint, Sid2p may well be blocking the onset of ring constriction by its stabilization under cell wall stress, even with a competent GS.

In animal cells, RhoA and RhoGEF Ect2p are key players in cytokinesis. Active Rho A around the equator regulates formins that polymerize actin filaments and kinases that activate myosin-II (*Wagner and Glotzer, 2016*). In addition, RhoA has a role as a negative regulator of the hippo pathway (the SIN counterpart in mammals) (*Plouffe et al., 2016*). When deleted RHOA, LATS1/2 (Sid2p counterpart in mammals), and YAP/TAZ (the transcription factor downstream) remained highly phosphorylated; the latest is sequestered in the cytoplasm, and is maintained transcriptionally inactive, even in the presence of serum (*Plouffe et al., 2016*). According to that, our work in *S. pombe* indicates that Rho1p controls the SIN to achieve ring constriction properly under cell wall stress. Understanding the relationship between Rho1p and the SIN pathway in yeasts would be helpful to grasps how RhoA and the Hippo pathway interact in mammals.

# Materials and methods

**Key resources table**

| Reagent type (species) or resource | Designation | Source or reference | Identifiers | Additional information |
|---|---|---|---|---|
| Antibody | Anti-phospho-p42/44 (Rabbit polyclonal) | Cell Signaling | Cat#: 9101, RRID:AB_331646 | WB (1:2500) |
| Antibody | Anti-HA (Mouse monoclonal) | Roche | Cat# 11666606001, RRID:AB_514506 | WB (1:5000) |
| Antibody | HRP anti-mouse (Goat polyclonal) | Bio-Rad | Cat#: 170–6516, RRID:AB_11125547 | WB (1:10000) |
| Antibody | HRP anti-rabbit (Goat polyclonal) | Bio-Rad | Cat#: 170–5046, RRID:AB_11125757 | WB (1:15000) |
| Chemical compound, drug | Blankophor | Bayer | Blankophor BA 267% | |
| Chemical compound, drug | KCl | Merck | Cat#: 104936 | |
| Chemical compound, drug | Caffeine | Sigma-Aldrich | Cat#: W222402 | |
| Chemical compound, drug | Caspofungin | Sigma-Aldrich | Cat#: SML0425 | |
| Chemical compound, drug | Latrunculin A | Sigma-Aldrich | Cat#: L5163 | |
| Chemical compound, drug | Hydroxyurea | Sigma-Aldrich | Cat#: H8627 | |
| Chemical compound, drug | Soybean lectin | Sigma Aldrich | Cat#: L2650 | |
| Commercial assay, kit | Ni-NTA | Novagen | Cat#: 70666 | |
| Software, algorithm | GraphPad Prism | GraphPad Prism (https://graphpad.com) | RRID:SCR_015807 | |
| Software, algorithm | ImageJ | ImageJ (http://imagej.nih.gov/ij/) | RRID:SCR_003070 | |
| Other | μ-Slide eight well | Ibidi | Cat#: 80826 | |

## General yeast growth conditions and genetic methods

Standard *S. pombe* genetic procedures and media were used (*Forsburg and Rhind, 2006*; *Moreno et al., 1991*). The relevant genotypes and the source of the strains used are listed in *Table 3*. Unless otherwise stated, the experiments were performed with cells growing exponentially in liquid-rich medium, yeast extract with supplements (YES; 0.5% yeast extract, 3% glucose, 225 mg/l adenine sulphate, histidine, leucine, uracil and lysine, 2% agar) and incubated at 28°C. Geneticin, hygromycin and nourseothricin were used at 120 µg/ml, 400 µg/ml, and 50 µg/ml, respectively. Latrunculin A (stock at 5 mM in DMSO) was used at 5 µM. To Tag Rgf1p with the GFP$^{Envy}$ protein we made used of the integrative plasmid pGR49 (pJK148- *rgf1*$^+$-GFP) carrying the GFP$^{Envy}$ in frame in a *Not*I site just before the TAA stop codon (*García et al., 2006a*). The GFP$^{Envy}$ was PCR-amplified from plasmid pFA6A-link-envy-SpHis5 (*Slubowski et al., 2015*) with NotI sites at the ends and sub-cloned into pGR41 instead of the GFP. This plasmid was cut with *Eco47*III and integrated into the *leu1* locus of an *rgf1Δ* strain. Genetic crosses and selection of the characters of interest by random spore analysis were used to combine different traits (*Forsburg and Rhind, 2006*). All the deletions strains generated in this work were checked by PCR, drop assays with caspofungin (0,5–1 µg/ml) and the appropriate selection markers.

## Microscopy and image analysis

Fluorescence images were acquired on an Olympus IX71 inverted microscope (Olympus) equipped with a PlanApo 100x/1.40 IX70 oil immersion objective, a personal DeltaVision system (GE Healthcare) and a CoolSnap HQ2 camera (Photometrics), all controlled by SoftWoRx Resolve 3D (Applied Precision). Early logarithmic phase cell cultures grown at 25°C were shifted to 36°C for 3 hr. Approximately 100 µl of cells were collected by a short spin, dissolved in 5 µl of water and Calfofluor white (Blankophor BA 267%, Bayer) was added at a final concentration of 50 µg/ml. Then, 10 slices at 0.2 µm were taken, corrected by 3D Deconvolution (conservative ratio, 10 iterations and medium noise filtering) with softWoRx software (GE Healthcare), and processed with Fiji distribution of Image

**Table 3.** *S. pombe* strains used in this work.

| Strains | Genotypes |
|---|---|
| YS5261 | h⁻*rgf1::kanMX6, leu1-32::rgf1-EnvyGFP, sid4-mcherry:hph, rlc1-tdTomato:nat, ura4D18* |
| SM440 | h⁺ *leu1-32::rgf1-GFP:leu1⁺, nda3-KM311, rgf1::nat, leu1-32* |
| TE519 | h⁻ *cdc15-140, rlc1-tdTomato:natMX6, rgf1::his3⁺, leu1-32::rgf1-GFP:leu1⁺* |
| TE348 | h⁻*cdc11-119, rlc1-tdTomato:natMX6, rgf1::kanMX6, leu1-32::rgf1-GFP:leu1⁺* |
| PG40 | h⁻ *rgf1::his3⁺, his3D1, ura4D18, leu1-32::rgf1-GFP:leu1⁺, ade6M210* |
| NG319 | h⁺ *cdc3-6, rgf1::his3⁺, his3D1, leu1-32::rgf1-GFP:leu1⁺* |
| TE149 | h⁻*sid2-250, leu1-32::rgf1-GFP:leu1⁺, rgf1::natMX6* |
| SM213[a] | h⁺ *leu1-32, ura4d18* |
| SM341 | h⁺ *rgf1::natMX6, leu1-32, ura4D18* |
| YS864 | h⁺ *cdc4-8, leu1-32* |
| TE377 | h⁺ *cdc4-8, rgf1::kanMX6, leu1-32* |
| YS862 | h⁻*rlc1::kanMX6, ura4D18, leu1-32, ade6M210* |
| TE389 | h⁺ *rlc1::kanMX6, rgf1::nat, ura4D18, leu1-32, ade6M210* |
| YS586[b] | h⁻*cdc15-140, leu1-32* |
| NG203 | h⁻*cdc15-140, rgf1::kanMX6* |
| TE246 | h⁺*imp2::ura4+, leu1-32, ade6M210* |
| TE454 | h⁺ *imp2::ura4⁺, rgf1::natMX6, leu1-32* |
| YS865 | h⁺ *cdc12-112* |
| TE251 | h⁻ *cdc12-112, rgf1::natMX6* |
| TE478 | h⁻*imp2-GFP:kanMX6, leu1:sid4-GFP, ura4D18, leu1-32* |
| TE495 | h⁻*rgf1::natMX6, imp2-GFP:kanMX6, leu1:sid4-GFP, ura4D18, leu1-32* |
| TE249[c] | h⁺ *leu1::GFP-atb2:ura4⁺, rlc1-tdTomato:natMX6, leu1-32, ura4D18, his3D1* |
| TE257 | h⁺*leu1::GFP-atb2:ura4⁺, rlc1-tdTomato:natMX6, rgf1::kanMX6, leu1-32, ura4D18* |
| TE470 | h⁺*LactC2-GFP:natMX6, sad1-GFP:kanMX6, leu1-32, ura4D18, ade6M210* |
| TE472 | h⁺*LactC2-GFP:natMX6, sad1-GFP:kanMX6, rgf1::natMX6, leu1-32, ura4D18, ade6M210* |
| TE399 | h⁺ *mEGFP-myo2:kanMX6, sid4-mCherry:hph, leu1-32, ura4D18* |
| TE402 | h⁺ *mEGFP-myo2:kanMX6, sid4-mCherry:hph, rgf1::natMX6, leu1-32, ura4D18* |
| TE450 | h⁺ *sad1-GFP:kanMX6, imp2-GFP:kanMX6, ura4D18* |
| TE452 | h⁺*sad1-GFP:kanMX6, imp2-GFP:kanMX6, rgf1::natMX6, ura4D18* |
| TE491 | h⁻*pmk1::ura4⁺, imp2-GFP:kanMX6, leu1:sid4-GFP, ura4D18, leu1-32* |
| TE585 | h⁻*pmk1::ura4⁺, mEGFP-myo2:kanMX6, sid4-mcherry:hph, ura4D18* |
| TE545 | h⁻*sty1::ura4⁺, imp2-GFP:kanMX6, leu1:sid4-GFP, ura4D18, leu1-32* |
| TE500 | h⁻*pck2::kanMX6, imp2-GFP:kanMX6, leu1:sid4-GFP, ura4D18, leu1-32* |
| TE493 | h⁻*pck1::kanMX6, imp2-GFP:kanMX6, leu1:sid4-GFP, ura4D18, leu1-32* |
| TE418 | h⁻*rho2::natMX6, imp2-GFP:kanMX6, leu1:sid4-GFP, ura4D18, leu1-32* |
| TE580 | h⁻*rho1-596:natMX6, imp2-GFP:kanMX6, leu1:sid4-GFP, ura4D18, leu1-32* |
| RC34 | h⁻*leu1-32::rgf1-PTTR-GFP:leu1⁺, rlc1-tdTomato:natMX6, rgf1::kanMX6, sid4-mCherry:hph, leu1-32* |
| TE562 | h⁻*cwg1-1, imp2-GFP:kanMX6, leu1:sid4-GFP, ura4D18, leu1-32* |
| TE564 | h⁻*cwg1-1, pmk1::ura4⁺, imp2-GFP:kanMX6, leu1:sid4-GFP, ura4D18, leu1-32* |
| TE551[a] | h⁻ *pmk1::kanMX6, leu1-32, ura4D18, ade6M210* |

*Table 3 continued on next page*

Table 3 continued

| Strains | Genotypes |
| --- | --- |
| TE541 | h⁻ pmk1::ura4⁺, hht1-RFP:kanMX6, imp2-GFP:kanMX6, leu1:sid4-GFP, ura4D18 |
| TE552 | h⁻ myp2::ura4⁺, imp2-GFP:kanMX6, leu1:sid4-GFP, ura4D18, leu1-32 |
| TE615 | h⁻rlc1::kanMX6, imp2-GFP:kan, leu1:sid4-GFP, ura4D18, leu1-32 |
| TE513 | h⁺ cdc7-GFP:ura4⁺, sad1:dsRed:natMX6, ura4D18 |
| YS826 | h⁺ sid2-GFP:ura4⁺, leu1-32, ade6M210, ura4D18 |
| TE427 | h⁺ sid2-GFP:ura4⁺, rgf1::natMX6, leu1-32, ade6M210, ura4D18 |
| TE413 | h⁺ cdc7-GFP:ura4⁺, imp2-GFP:kanMX6 |
| TE414 | h⁺cdc7-GFP:ura4+, imp2-GFP:kanMX6, rgf1::natMX6 |
| TE527 | h⁺ clp1-GFP:kanMX6, imp2-GFP:kanMX6, leu1:sid4-GFP, leu1-32 |
| TE507 | h⁻ clp1::kanMX6, imp2-GFP:kanMX6, leu1:sid4-GFP, leu1-32 |
| EM352[d] | h⁻ pact1-LA:GFP:leu, ura4D18, leu1-32 |
| RC8 | h⁻sid2-250, imp2-GFP:kanMX6, sfi1-GFP:kanMX6 |
| TE611 | h⁻sid1-239, imp2-GFP:kanMX6, leu1:sid4-GFP, leu1-32 |
| YS5154 | h⁻adf1-1, imp2-GFP:kanMX6, leu1:sid4-GFP, ura4D18, leu1-32 |
| TE592 | h⁻cdc12-112, imp2-GFP:kanMX6, leu1:sid4-GFP, ura4D18, leu1-32 |
| TE591 | h⁻pmk1:: kanMX6, pmk1(K52E)−3 HA:leu1⁺, imp2-GFP:kanMX6, sfi1-GFP:kanMX6, leu1-32 |
| PG323 | h⁻pmk1-HA6h:ura4⁺, his3D1, leu1-32 |
| YS5432 | h⁻cwg1-1:natMX6, pmk1-HA6h:ura4⁺ |
| TE261[a] | h⁻cdc15-HA:kanMX6, ura4D18, leu1-32 |
| TE646 | h⁻pmk1::ura4⁺, cdc15-HA:kanMX6, ura4D18 |
| TE350 | h⁻mid1-GFP:kanMX6, rlc1-tdTomato:natMX6, leu1-32, ura4D18, his3D1 |
| TE353 | h⁻mid1-GFP:kanMX6, rlc1-tdTomato:natMX6, rgf1::his3⁺, leu1-32, ura4D18, his3D1 |
| TE587 | h⁻rgf1::natMX6, pmk1::ura4⁺, imp2-GFP:kanMX6, leu1:sid4-GFP, ura4D18, leu1-32 |

All strains were generated in this study except for strains with label[a] from P. Perez (IBFG, University of Salamanca), label[b] from H. Valdivieso (IBFG, University of Salamanca), label[c] from J.C. Ribas (IBFG, University of Salamanca), label[d] from M. Balasubramanian (University of Warwick).

(*Schindelin et al., 2012*). Super-resolution images were obtained using an Olympus IXplore SpinSR SoRa confocal Spinning disk microscope (Olympus), composed by the microscope Olympus IX83 (Olympus), U Plan Super Apochromatic 100x/1.45 oil immersion objective and sCMOS ORCA Flash 4.0 V3 camera (Hamamatsu).

Time-lapse imaging was carried out separately on two different microscopes: the previously described DeltaVision system and an Olympus IX81 spinning disk microscope (Olympus) equipped with a PlanApo 100x/1.40 oil immersion objective, a confocal CSUX1-A1 module (Yokogawa), and an Evolve (Photometrics) camera controlled by Metamorph software (Molecular Devices). One ml of midlog-phase cell cultures was collected by centrifugation (3500 g, 1 min) resuspended in 0.3 ml of YES containing the respective drugs and placed in a well of a μ-Slide eight well (Ibidi). Each of the wells was previously coated with 5 μl of 1 mg/ml soybean lectin (Sigma-Aldrich). Cells were allowed to adhere to the bottom of the well for 2 min before removing the media. Then the cells were washed three times with media and resuspended in 0.3 ml of the same media. Time-lapse experiments were performed at 28°C. For imaging the temperature-sensitive strains at higher temperatures, the cells were shifted from 25°C to the described temperature and incubated for 1 hr prior to imaging. Using the DeltaVision system, 7 slices at 0.6 μm were taken every 3 min within up to three different regions. With the spinning disk, 10 slices at 0.4 μm were taken every 3 min in up to four regions in each of the four different wells.

In all time-lapse series, we used the SPB separation as time zero of the cellular clock. We define cytokinesis completion time as the time between the SPB separation and the complete closure of the AR. The time of the three stages of the CAR ring are defined as follows: assembly, from the appearance of nodes until there is a uniform ring signal; maturation, from a uniform ring signal to the start of its change in diameter; and constriction, from the first diameter change to its complete closure as a single point.

Total fluorescence of Cdc7p-GFP over time was measured using the sum projection of the slices taken in the spinning disk system. An area empty of cells was used to determine the average background fluorescence that was subtracted from the images. We used Sad1p-DsRed signal as a marker of both SPBs to select them with a 5-pixel diameter circle (0.37 $\mu m^2$), and we measured the total intensity of the green channel in these regions. Clp1-GFP nuclear intensity over time was measured using the sum projection of the slices taken in the DeltaVision system. An area empty of cells was used to determine the average background fluorescence that was subtracted from the images. Then, Sid4p-GFP was used as a marker of the SPBs to locate the nucleus, encircling it within a 42-pixel diameter circle (6.14 $\mu m^2$), which contained Sid4p in its periphery. The total intensity of the green channel in these regions was subsequently measured. The data were normalized according to their maximum and minimum.

## Purification and detection of activated Pmk1

Cells from 30 ml of culture were harvested by centrifugation at 4°C, 500 g for 3 min, washed with cold PBS buffer, and immediately frozen in liquid nitrogen. Cell homogenates were prepared under native conditions employing chilled acid-washed glass beads and lysis buffer (10% glycerol, 50 mM Tris-HCl, pH 7.5, 150 mM NaCl, 0.1% Nonidet NP-40, plus a specific protease inhibitor cocktail: 100 M $p$-aminophenyl methanesulfonyl fluoride, leupeptin, and aprotinin). The lysates were cleared by centrifugation at 13,000 rpm for 15 min and Pmk1-HA6H was purified with Ni2-NTA-agarose beads (Novagen). The purified proteins were loaded on 10% SDS-PAGE gels, transferred to an Immobilon-P membrane (Millipore), and incubated with either monoclonal mouse anti-HA (12CA5, Roche) or polyclonal rabbit anti-phospho-p42/44 antibodies (Cell Signaling). The immunoreactive bands were revealed with anti-mouse or anti-rabbit HRP secondary antibodies (Bio-Rad) and the ECL detection kit (Amersham Biosciences).

## Western blot of synchronized cell cultures

Early-log phase cultures of cells containing Cdc15-HA were synchronized in S phase using hydroxyurea (HU) at a final concentration of 12.5 mM. After 4 hr of growth at 32°C HU was washed out and the cell cultures were released at 28°C, treated or untreated with BP (1 mg/ml). Samples were collected every 30 min after the first hour. 10 ml of cultures were mixed with 40 ml of 50 mM Tris-HCl pH 7. Cells were immediately harvested by centrifugation at 4°C, 500 g for 3 min and frozen at –80°C in 100 ml of 20% TCA. Protein extracts were obtained by TCA precipitation. Proteins were resolved with 7% SDS-PAGE gels, transferred to an Immobilon-P membrane (Millipore), incubated with monoclonal mouse anti-HA antibodies (12CA5, Roche) and revealed with anti-mouse HRP secondary antibodies (Bio-Rad) and the ECL detection kit (Amersham Biosciences).

## Acknowledgements

We wish to thank Cesar Roncero and Henar Valdivieso for their support and our colleges from the Cell Wall group of the Institute of Functional Biology and Genomics at the University Salamanca. Special thanks to Sergio Rincon for his very helpful comments on the manuscript and Carmen Castro for help with microscopy. The English text was revised by E Keck. T Edreira and E Manjon were financially supported by a contract obtained through the Regional Government of Castile and Leon, co-financed by the European Social Fund. R Celador was financially supported by a collaboration grant from the University of Salamanca. Funding sources: MEIC, Spain (BFU2017-84508-P; Regional Government of Castile and Leon [SA073U14]). Funding for open access charge: MEIC.

## Additional information

### Funding

| Funder | Grant reference number | Author |
|---|---|---|
| Regional Government of Castile and Leon | The European Social Fund | Tomás Edreira<br>Elvira Manjón |
| University of Salamanca | | Rubén Celador |

The funders had no role in study design, data collection and interpretation, or the decision to submit the work for publication.

### Author contributions
Tomás Edreira, Conceptualization, Formal analysis, Validation, Investigation, Methodology, Writing - review and editing; Rubén Celador, Conceptualization, Formal analysis, Validation, Investigation, Methodology; Elvira Manjón, Supervision, Validation, Investigation, Methodology; Yolanda Sánchez, Conceptualization, Formal analysis, Supervision, Funding acquisition, Investigation, Writing - original draft, Writing - review and editing

### Author ORCIDs
Tomás Edreira (iD) https://orcid.org/0000-0002-1985-0940
Yolanda Sánchez (iD) https://orcid.org/0000-0002-7650-9319

### Decision letter and Author response
Decision letter https://doi.org/10.7554/eLife.59333.sa1
Author response https://doi.org/10.7554/eLife.59333.sa2

## Additional files

### Supplementary files
• Transparent reporting form

### Data availability
All data generated or analysed during this study are included in the manuscript and supporting files.

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
