## [Decision Letter]

**Acceptance summary:**

This manuscript explores the mechanism of initiation of contraction of the cytokinetic actomyosin ring and how the initiation step is regulated to ensure maximum cell viability. In particular, the authors find that cell wall stress causes a delay in actomyosin ring contraction and this delay is dependent on the kinase Pmk1 and GTPase signaling pathways. The work combines some excellent microscopy of the cytokinetic apparatus in cell wall stressed wild-type and pmk1 or rgf1 mutants in a variety of conditions to unveil a new regulatory network regulating genome stability and cell viability.

**Decision letter after peer review:**

Thank you for submitting your article "A novel checkpoint pathway controls actomyosin ring constriction trigger in fission yeast" for consideration by *eLife*. Your article has been reviewed by three peer reviewers, one of whom is a member of our Board of Reviewing Editors, and the evaluation has been overseen by Anna Akhmanova as the Senior Editor. The following individual involved in review of your submission has agreed to reveal their identity: Masayuki Onishi (Reviewer #3).

The reviewers have discussed the reviews with one another and the Reviewing Editor has drafted this decision to help you prepare a revised submission.

The reviewers and the editors agreed that your work is breaking new ground into understanding the mechanism of the septation checkpoint and the unexpected links of the BP induced ring contraction delay to the Pmk1 pathway and the Rho-GTPase pathways. The reviewers also found that a few additional experiments will strengthen the work before publication.

Below I have summarized the essential revisions for this work. I have also included the reviewer comments verbatim, since they contain a large number of excellent suggestions to carry the work forward in future studies.

1) rgf1∆ pmk1∆ double mutants. I think it is critical that you repeat (at the very least) the experiments in Figure 4 in this double-mutant background.

2) A concern of two reviewers centers around the question of the reason and validity of using Imp2 as the marker for CAR in rfg1∆ cells. This point clearly needs to be addressed.

3) Two reviewers raised a question about the band-shift data. This clearly needs to be addressed.

4) Two reviewers (reviewer 3 point 1 and reviewer 2 point 7) raise a concern about the putative connection between the CIP and septum structure / cell separation defects. It will be terrific if it was possible to carry out some EM work of the septum architecture (you may already have the data as well). In the absence of this the Discussion should be more hypothetical and balanced.

5) Reviewer 1 raised a question about Mid1 localization (point 3). This should be done as it strengthens the main point of this manuscript that the CIP delays the onset of CAR constriction.

Reviewer #1:

The findings and conclusions in the paper (Edreira et al.,) are very well supported by the experiments. Overall, this paper uses a range of techniques (live cell imaging, genetics and biochemistry) to elegantly elucidate how cell wall integrity pathway halts cytokinesis upon cell stress through SIN pathway. It is an interesting paper, clearly written and largely convincing. The authors should hence address the following major points to improve the manuscript prior to acceptance. This will improve the rigor of the work. Of course, there may be constraints due to the covid19 situation that prevent many experiments.

1) The authors should look at the localisation (ring proteins are also carrying membrane anchoring domains) of Sbg1 (Sethi et al., 2016; Davidson et al., 2016) and Blt1 in wild type and ∆rgf1 cells with and without BP.

2) The authors should look at various ring localising proteins (relevant to assembly and constriction) in ∆rgf1 mutants with and without BP.

3) Anilin like protein Mid1 leaves the ring prior to ring constriction.

The authors should look at mid1 localisation (disappearance from the ring) during the onset of ring constriction in wild type and ∆rgf1 cells with and without BP and check if mid1 stays longer or leaves the ring earlier in comparison to WT cells.

4) Is there any specific reason why the authors have used Imp2-GFP as a ring marker and not a conventional marker (rlc1 or myo2)? Is it due to genetic interactions of ring markers with ∆rgf1 cells ?

5) The authors should test the Rlc1-S35A and S36A mutants and/or shk1 mutants (Loo et al., JCB 2010) to see the constriction timings in combination with ∆rgf1 in the presence and absence of BP.

6) Did the authors look at the polarisome components upon BP treatment in ∆rgf1 background cells. Polarity kinases (pom1) phosphorylates some of the key components of the rings (i.e Cdc15)

7) Are there any direct interaction of SIN components with Rgf1 at the cell division site?

8) Upon Cell wall damage/stress the rings seems to be more stable, did the authors look at the dynamics of actin bundling proteins (i.e alpha actinin) at the cell division site using the same conditions as above.

9) The Authors should provide some evidence or discuss how Rho1 could possibly control SIN activation via Sid2 kinase activity.

Is there any downstream components of Rho1 pathway controlling timely activation of SIN via Sid2/Mob1?

Reviewer #2:

The authors found that fission yeast cells delay onset of cytokinetic ring constriction upon cell wall defects. The impact of this study is high because the authors define two signaling pathways that appear to mediate this novel checkpoint response. By defining the pathways, the authors demonstrate that the checkpoint delay occurs due to a regulatable response. The authors have not identified the “sensor” for the checkpoint, and several connections within the signaling pathways remain unknown. However, these limitations are offset by the large amount of interesting data that establish this new checkpoint and set the framework to understand the underlying signaling mechanisms. The two signaling pathways (MAPK and SIN) that relay this signal are conserved in other organisms, so similar mechanisms might operate broadly in similar or related ways even in cell types that do not have a cell wall.

1) The authors demonstrate that addition of BP delays onset of CAR constriction. This result is important for interpretation of past studies that have used BP (or CW) to monitor the timing of cytokinesis events (e.g. Cortes et al., 2018; Ramos et al., 2019). The authors might discuss the implication of their results for past studies.

2) It is unclear to me if BP-treated cells detect a defect in the cell wall organization, or alternatively if they experience an imbalance in force production between the CAR and septum. If force generation by the septum is reduced by BP, then the CAR might need to provide additional force. I don't see any data in the paper that make a strong argument for one possibility versus the other. I don't think that the authors need to resolve this question in the current study, but they might alter their interpretations to include the possibility that Pmk1-SIN signaling delays CAR constriction so that the CAR can recruit additional force-generating machinery.

3) The mechanism connecting activated Pmk1 with SIN signaling remains unknown. At first glance, the timing of this checkpoint suggests that it operates too rapidly to involve changes gene expression. Instead, it seems possible that Pmk1 phosphorylates component(s) of the SIN pathway. Have the authors looked for Pmk1 consensus sites in SIN proteins, or have any SIN components been identified at Pmk1 targets? Some additional consideration of the underlying mechanism would strengthen the framework for future studies.

4) Some additional analysis of double mutants would strengthen the conclusions regarding the pathway that signals to Pmk1. For example in Figure 4, I expected to see the phenotype for pmk1∆ rgf1∆ double mutants, with the prediction that they do not exhibit an additive phenotype on the checkpoint.

5) The authors' experiments build a signaling pathway that leads to activating phosphorylation of Pmk1, but the pathway can be equally impacted by modulating the phosphatase Pmp1, which removes the activating phosphorylation on Pmk1. It does not seem essential for the authors to test the identity or regulation of CIP phosphatases, but they should discuss potential regulatory mechanisms in the text. It seems possible that BP or other stresses might reduce Pmp1 activity, which would facilitate increased activating phosphorylation of Pmk1 by lowering the inhibitory threshold.

6) Figure 5B could be improved by adding more labels to distinguish WT versus pmk1∆ data.

7) The connection between the checkpoint pathway and the septum defect in Figure 5 remains murky. The authors suggest that the checkpoint induces "the quality control machinery for septum assembly and/or disassembly." I am not aware of evidence that these cells build "a defective septum that was difficult to digest by the glucanases." This speculation is consistent with their data but should be phrased much more hypothetically.

8) From experiments combining myp2∆ with BP, the authors conclude that "there could be a synergy between the cell wall stress and Myp2 motor action." I recommend that the authors reword this interpretation to explain that Myp2 and BP act independently on the timing of cytokinesis, as shown by the additive defect of myp2∆ and BP.

9) The authors conclude that the CAR is more stable upon BP treatment based on low-dose LatA experiments. A more direct test of CAR stability would be the timing of disassembly upon high-dose LatA treatment. The two tests (high and low dose) are complementary and would provide a stronger result in combination. The low-dose experiment sets up a competition between CAR and other actin structures, as opposed to directly testing how quickly the CAR actin turns over.

10) I do not find the band shift experiments in Figure 6E convincing. One possibility is to run all of the 120 timepoint samples next to each other. As presented, the subtle band shift leaves comparison of different samples quite challenging. This figure also needs a phosphatase-treated control to establish the amount of band shift in the authors' system.

11) The authors use fixed timepoint microscopy in Figure 7A to conclude that "ring disassembly dynamics was similar in the presence of absence of BP" for sid2-250 mutants. It seems important to perform timelapse imaging and to measure disassembly dyamics for individual rings, if the authors want to draw this conclusion.

12) Figure 7B provides a very convincing quantification of Cdc7 dynamic localization at SPBs. I would recommend that the authors present a similar graph for the data in Figure 7—figure supplement 1.

Reviewer #3:

The Rho-GTPase pathway has been implicated in cytokinesis with roles in regulation of the contractile actomyosin ring (CAR), cell-wall synthesis (in budding and fission yeasts), and other signaling pathways. In this paper, the authors showed that the Rho1 GEF Rfg1 localizes in the region following the contractile ring during cell division, in a CAR-dependent manner. Interestingly, the authors found that treatment of cells with the BP dye that specifically binds to linear 1,3-β-glucan delays cytokinesis in WT, but not in rfg1∆ cells. The delay was largely in the timing of onset of CAR constriction and concomitant ingression of plasma membrane, rather than the rate of constriction/ingression. Based on these results, the authors hypothesize that Rfg1 is involved in a mechanism that senses cell-wall perturbation and delays the onset of cytokinesis. Because Rfg1 has previously been implicated in the cell-wall integrity pathway (CIP), the authors tested the requirement of known CIP components for BP-responsive cytokinesis delay, and identified Rho1p, Pck2p, and Pmk1p as downstream factors of Rfg1p in this response. pmk1∆ cells show significant increase in septation index upon BP treatment, which the authors conclude to be a consequence of failure to delay cytokinesis onset. The delay in the onset of cytokinesis correlated with prolonged cytoplasmic retention of the Clp1 phosphatase and dephosphorylation of its substrate Cdc15. Finally, the delay in cytokinesis onset requires normal SIN activity, implicating the pathway in this process.

Contribution of cell wall to cytokinesis has been clearly demonstrated in both fission yeast and budding yeast, yet its regulation and coordination with the CAR pathway has not been fully established. This paper provides a new insight into the ability of the cells of *S. pombe* in controlling CAR constriction, septum synthesis, and membrane ingression in response to perturbation to cell-wall synthesis. Most of the experiments are performed carefully and data are clearly presented, which are of interest to the readers in the field.

1) While the manuscript clearly demonstrates the cell's ability to respond to cell-wall perturbation that involves delayed onset of CAR constriction and septum synthesis, the biological significance of this observation is unclear. How did *S. pombe* evolve to have this adaptation pathway, and what is the consequence of a defect in it? Although the authors conclude that the observed separation defect is a consequence of the failure to delay cytokinesis onset, the causality between the two phenotypes does not seem to be fully established. The separation defect could be a separate consequence of the defective CIP under cell-wall stress, such as through some defects in the secretory pathway, that is independent of the delayed cytokinesis onset. Do any of the other CIP pathway mutants show separation defects, and if they do, is there a good correlation between the timing of Imp2 constriction (Figure 4) and separation defects? Do mutations in myp2 etc. that delay the onset of constriction also rescue the separation defect in pmk1∆? Can the authors speculate as to whether the cell-separation defect causes a severe morphological or ecological disadvantage that explains the evolution of this response pathway?

2) In many figures, rgf1∆ and pmk1∆ are used interchangeably to test the role of the proposed pathway in delaying cytokinesis onset. While this is consistent with the previously established CIP pathway, in which Rgf1 and Pmk1 are positioned at the most upstream and downstream (Figure 4B), it is possible that the two proteins function independently in separate or partially overlapping pathways leading to cytokinesis delay in BP-treated cells. Thus, the authors should at least establish that rgf1∆ and pmk1∆ mutations do not have any additive effects in the BP response, and ideally repeat some key experiments in both mutant backgrounds.

---

## [Author Response]

Below I have summarized the essential revisions for this work. I have also included the reviewer comments verbatim, since they contain a large number of excellent suggestions to carry the work forward in future studies.1) rgf1∆ pmk1∆ double mutants. I think it is critical that you repeat (at the very least) the experiments in Figure 4 in this double-mutant background.

We have included results that show *rgf1*∆ and *pmk1*∆ mutations do not have, as expected, additive effects in the BP response (Figure 4—figure supplement 4).

The following sentence has been included:

“Moreover, an *rgf1*Δ *pmk1*Δ double mutant exhibit no synergistic effects in the BP response (Figure 4—figure supplement 4), suggesting that the Rgf1p branch activating Pmk1p (Figure 4B) channels an efficient cytokinesis delay….”

2) A concern of two reviewers centers around the question of the reason and validity of using Imp2 as the marker for CAR in rfg1∆ cells. This point clearly needs to be addressed.

We have used different markers to validate the delay seen in wild type cells compared to *rgf1*Δ. Imp2p-GFP (Figure 2A), Rlc1p-tdTomato (Figure 2—figure supplement 1), mEGFP-Myo2p (Figure 3A and Figure 4—figure supplement 1), all of them behave similarly. Regarding the genetic interactions, while double mutants do not thrive above certain temperatures, the fluorescent-tagged versions grow in both wild type and *rgf1*Δ backgrounds, from 25ºC to 37ºC. We used mostly Imp2p-GFP instead of mEGFP-Myo2p and Rlc1p-tdTomato because Imp2p is first seen to the CAR long after SPBs separation. Since we are interested in ring maturation/constriction, the use of Imp2p-GFP allows us to visualize both structures (CAR and SPBs) within the same fluorescent channel, thus minimizing photodamage, as opposed to other ring components that form nodes.

3) Two reviewers raised a question about the band-shift data. This clearly needs to be addressed.

We have tested different acry/bis gel percentage and running conditions trying to get a better band shift. Moreover, we have repeated the experiment at least 4 times with similar results (see Author response image 1). In the presence of BP, the samples of 90 and 120 minutes always show slightly higher mobility indicative of a sort of “hipophosphorylated form”. However, in wild type (w/o BP) and *pmk1*Δ (+BP) cells the 120-minute sample is always running a little bit slower than the 90 minutes sample (see the yellow line on the right panels). It is probable that 2h after release most cells have completed septation and carry Cdc15p in a more phosphorylated form. The replicas are shown in the supplemental material (Figure 6—figure supplement 3).

As suggested by the reviewer 3, the 120 time point samples were run next to each other (see Author response image 1)

4) Two reviewers (reviewer 3 point 1 and reviewer 2 point 7) raise a concern about the putative connection between the CIP and septum structure / cell separation defects. It will be terrific if it was possible to carry out some EM work of the septum architecture (you may already have the data as well). In the absence of this the Discussion should be more hypothetical and balanced.

For this part of the Discussion, we have built on previous results. It is known that under conditions of nutrient limitation, hypertonic stress or caffeine, *pmk1*∆ cells grow as short-branched filaments in which the cell walls and septa are thickened (Zaitsevskaya-Carter, 1997 #982). Apparently, these cells had almost finished cytokinesis, but had not completely lysed the external wall (Zaitsevskaya-Carter, 1997 #982).

We have included the following sentence in the Discussion:

“In the case that the checkpoint is overridden, some septa could be poorly built up. In that sense it is known that conditions of nutrient limitation, hypertonic stress or caffeine, prompt *pmk1*∆ cells to grow as short-branched filaments in which the cell walls and septa are thickened (Zaitsevskaya-Carter and Cooper, 1997). At the same time, Pmk1p might induce the expression of genes that function to replace the damaged wall and restart cytokinesis even under stress conditions (D. Chen et al., 2003; Takada et al., 2007).”

5) Reviewer 1 raised a question about Mid1 localization (point 3). This should be done as it strengthens the main point of this manuscript that the CIP delays the onset of CAR constriction.

In the revised version we have included kymographs and quantifications to show the disappearance of Mid1p-GFP from the ring in wild type and *rgf1*Δ cells treated with BP.

“To analyze the onset and the rate of constriction in more detail, first we monitored the time taken for Mid1p-mEGFP to be cleared from the CAR in *rgf1*^+^ and *rgf1*Δ cells treated with BP. Mid1p–mEGFP dissociates from contractile rings during the maturation period (Sohrmann et al., 1996). According to the above results, Mid1p–mEGFP dissociated from contractile rings in 16.9±3.1 min in *rgf1*^+^, by contrast the protein dissociated prematurely from the ring in *rgf1*Δ cells (12.0±1.9 min) (Figure 3—figure supplement 1).”

Reviewer #1:The findings and conclusions in the paper (Edreira et al.,) are very well supported by the experiments. Overall, this paper uses a range of techniques (live cell imaging, genetics and biochemistry) to elegantly elucidate how cell wall integrity pathway halts cytokinesis upon cell stress through SIN pathway. It is an interesting paper, clearly written and largely convincing. The authors should hence address the following major points to improve the manuscript prior to acceptance. This will improve the rigor of the work. Of course, there may be constraints due to the covid19 situation that prevent many experiments.1) The authors should look at the localisation (ring proteins are also carrying membrane anchoring domains) of Sbg1 (Sethi et al., 2016; Davidson et al., 2016) and Blt1 in wild type and ∆rgf1 cells with and without BP.

Blt1p-GFP localizes very much alike in wild type and *rgf1*Δ cells with or w/o BP (see Author response image 2). Blt1p-GFP accumulated in interphase cortical nodes and remained in the ring during ring constriction and later until completion of septation in both strains with or w/o BP.

**Author response image 2. respfig2:** 

2) The authors should look at various ring localising proteins (relevant to assembly and constriction) in ∆rgf1 mutants with and without BP.

We have already checked in the paper the localization of several proteins important for CAR assembly Myo2p-EGFP (Figure 3 and Figure 4—figure supplement 1) and Rlc1p-tdTom (Figure 2—figure supplement 1) as well as proteins that play a role during constriction such as Cdc7p-GFP (Figure 7B and Figure 7—figure supplement 1) and Sid2p-GFP (Figure 7—figure supplement 1).

As suggested by the reviewer we have obtained new strains to analyze Myp2p-GFP (myosin II heavy chain), Ppb1p-GFP (calcium-dependent serine/threonine protein phosphatase calcineurin) and Cdc15p-GFP in wild type and *rgf1*Δ cells with or w/o BP (see response to reviewer 3).

Apparently, there are not significant differences in localization of Myp2p-GFP and Ppb1p-GFP between wild type and *rgf1*Δ cells (see Author response image 3); however, it will be necessary to analyze their localization during ring constriction by time-lapse microscopy.

**Author response image 3. respfig3:** 

3) Anilin like protein Mid1 leaves the ring prior to ring constriction.The authors should look at mid1 localisation (disappearance from the ring) during the onset of ring constriction in wild type and ∆rgf1 cells with and without BP and check if mid1 stays longer or leaves the ring earlier in comparison to WT cells.

In the revised version, we have included kymographs to show the disappearance of Mid1p-GFP from the ring in wild type and *rgf1*Δ cells treated with BP.

“To analyze the onset and the rate of constriction in more detail, first we monitored the time taken for Mid1p-mEGFP to be cleared from the CAR in *rgf1*^+^ and *rgf1*Δ cells treated with BP. Mid1p–mEGFP dissociates from contractile rings during the maturation period (Sohrmann et al., 1996). According to the above results, Mid1p–mEGFP dissociated from contractile rings in 16.9±3.1 min in *rgf1*^+^, by contrast the protein dissociated prematurely from the ring in *rgf1*Δ cells (12.0±1.9 min) (Figure 3—figure supplement 1).”

4) Is there any specific reason why the authors have used Imp2-GFP as a ring marker and not a conventional marker (rlc1 or myo2)? Is it due to genetic interactions of ring markers with ∆rgf1 cells ?

We have used different markers to validate the delay seen in wild type cells compared to *rgf1*Δ. Imp2p-GFP (Figure 2A), Rlc1p-tdTomato (Figure 2—figure supplement 1), mEGFP-Myo2p (Figure 3A and Figure 4—figure supplement1), all of them behave similarly. Regarding the genetic interactions, while double mutants do not thrive above certain temperatures, the fluorescent-tagged versions grow in both wild type and *rgf1*Δ backgrounds, from 25ºC to 37ºC. We used mostly Imp2p-GFP instead of mEGFP-Myo2p and Rlc1p-tdTomato because Imp2p is first seen to the CAR long after SPBs separation. Since we are interested in ring maturation/constriction, the use of Imp2p-GFP allows us to visualize both structures (CAR and SPBs) within the same fluorescent channel, thus minimizing photodamage, as opposed to other ring components that form nodes.

5) The authors should test the Rlc1-S35A and S36A mutants and/or shk1 mutants (Loo et al., JCB 2010) to see the constriction timings in combination with ∆rgf1 in the presence and absence of BP.

Because the Pak1p/Shk1p mutant cells also show accelerated cytokinesis, we had already checked our results against the work of Loo and Balasubramanian (Loo, 2008 #1930). Pak1p by phosphorylating Rlc1p function to delay CAR constriction and septum assembly until completion of chromosome segregation. The physiological consequences of the lack of Pak1p are shown in the *nmt1-pak1 ase1*Δ *d*ouble mutant. These cells displayed fully formed septa before full segregation of the chromosomes, some cells show the nucleus displaced to one compartment and in others septation takes place through the segregating nuclei.

We have not pursued this line of investigation because we haven’t seen a “cut-like phenotype” or anucleated cells in *ase1*Δ*rgf1*Δ *c*ells expressing Rlc1p-tdTomato and GFP-atb2 (unpublished results), nor in *rgf1*Δ or *pmk1*Δ mutants in the presence of cell wall damaging drugs (Figure 5A). Accordingly, while the initiation of ring constriction seemed to be earlier in *pak1* mutant cells, we haven’t seen differences in the initiation timing among *rgf1*Δ and wild-type cells (Figure 3).

6) Did the authors look at the polarisome components upon BP treatment in ∆rgf1 background cells. Polarity kinases (pom1) phosphorylates some of the key components of the rings (i.e Cdc15)

This work has been focused mainly on the ring dynamics. As reported previously, Pom1p-GFP was clearly observed at both cell tips in *rgf1*^+^ cells. In contrast, in our hands most *rgf1*Δ cells concentrated Pom1p-GFP signal at one end (the non-growing end, see Author response image 4).

**Author response image 4. respfig4:** 

Regarding the localization of Tea4p-GFP in the *rgf1*Δ background, there were no substantial differences in presence or absence of BP. However, the localization of Tea1p and Tea4p along the microtubules is better seen in the *rgf1*Δ background (see Author response image 5).

**Author response image 5. respfig5:** 

7) Are there any direct interaction of SIN components with Rgf1 at the cell division site?

There is not a direct interaction described to date, at least that we are aware off.

8) Upon Cell wall damage/stress the rings seems to be more stable, did the authors look at the dynamics of actin bundling proteins (i.e alpha actinin) at the cell division site using the same conditions as above.

Ain1p-GFP fluorescent signal is weak; however, the dynamics of Ain1p-GFP is not different that the other ring proteins that we have analyze before. Addition of BP onto wild-type cells caused a delay in the initiation of ring constriction marked by Ain1p-GFP.

**Author response image 6. respfig6:** 

9) The Authors should provide some evidence or discuss how Rho1 could possibly control SIN activation via Sid2 kinase activity.Is there any downstream components of Rho1 pathway controlling timely activation of SIN via Sid2/Mob1?

The relation of Rho1p and the SIN is not straightforward. Rho1p is essential to feedback activate Spg1p during actomyosin ring constriction acting through the SIN regulator Edt1p (Alcaide-Gavilan, 2014 #3113). However, to date there is no direct interaction among downstream components of the Rho1p pathway and Sid2p. It is possible that Pmk1p substrates or Pmk1p directly phosphorylate Sid2p. In that sense, Sid2p carries several kinase docking motifs (R/K)xxxx#x# (where # is a hydrophobic residue) of potential interaction with the ERK1/2 and p38 subfamilies of MAP kinases (Pmk1 orthologues).

Reviewer #2:The authors found that fission yeast cells delay onset of cytokinetic ring constriction upon cell wall defects. The impact of this study is high because the authors define two signaling pathways that appear to mediate this novel checkpoint response. By defining the pathways, the authors demonstrate that the checkpoint delay occurs due to a regulatable response. The authors have not identified the “sensor” for the checkpoint, and several connections within the signaling pathways remain unknown. However, these limitations are offset by the large amount of interesting data that establish this new checkpoint and set the framework to understand the underlying signaling mechanisms. The two signaling pathways (MAPK and SIN) that relay this signal are conserved in other organisms, so similar mechanisms might operate broadly in similar or related ways even in cell types that do not have a cell wall.1) The authors demonstrate that addition of BP delays onset of CAR constriction. This result is important for interpretation of past studies that have used BP (or CW) to monitor the timing of cytokinesis events (e.g. Cortes et al., 2018; Ramos et al., 2019). The authors might discuss the implication of their results for past studies.

The Calcofluor/Blankophor has been used in many laboratories, including ours, as a dye to visualize the cell periphery. Our results indicated that even very low concentrations of BP (5µg/ml) delays onset of CAR constriction slowing down cytokinesis. That said, as a dye BP stains the septum even at much lower concentrations, so the usage of slightly higher dilutions may have avoided this delay while still allowing septum monitoring.

2) It is unclear to me if BP-treated cells detect a defect in the cell wall organization, or alternatively if they experience an imbalance in force production between the CAR and septum. If force generation by the septum is reduced by BP, then the CAR might need to provide additional force. I don't see any data in the paper that make a strong argument for one possibility versus the other. I don't think that the authors need to resolve this question in the current study, but they might alter their interpretations to include the possibility that Pmk1-SIN signaling delays CAR constriction so that the CAR can recruit additional force-generating machinery.

In the first part of the question, both possibilities are not excluding. There may be an “imbalance in force production between the CAR and septum” due to the defect in the organization of the wall; it may even be that this imbalance is the way by which the defect is detected. We cannot differentiate one from the other.

The second part is about the additional force generated by the CAR and is not so easy to explain with our data. To begin with, we would have to assume that the CAR could exert sufficient constricting force over the septum to overcome the turgor pressure, something quite controversial. Second, we cannot explain the absence of delay in the *rgf1*Δ, *pmk1*Δ and *sid2-250* mutants based on the force generated by the CAR. Do they have a "stronger" ring? Do they recruit more components previously? Is their septum immune to loss of strength? Third, in the presence of cell wall damage there is no decrease in constriction speed while most mutants of ring components do contract but slower, which fits better with a lesser force generated from the CAR. The response here is more like to stop or not stop before constriction.

We have added a paragraph in the Discussion:

“Moreover, cell wall stress influences the onset of CAR constriction, but does not interfere with CAR assembly or the rate of CAR constriction. It is possible that in the temporary absence of a cell wall headway the CAR provides insufficient force to be closed. If BP reduces the force generated by the septum, then the cells could experience an imbalance in force production between the CAR and the septum and the CAR might need to provide additional force (Proctor, 2012 #5704). Alternatively, a regulatory pathway may couple constriction of the CAR and septum formation such that a defect in septum formation could prevent the constriction of the CAR. Some of these possibilities are not exclusive. We have shown that cell wall stress (inflicted by BP) generates a delay of approximately 15 min in CAR constriction”

3) The mechanism connecting activated Pmk1 with SIN signaling remains unknown. At first glance, the timing of this checkpoint suggests that it operates too rapidly to involve changes gene expression. Instead, it seems possible that Pmk1 phosphorylates component(s) of the SIN pathway. Have the authors looked for Pmk1 consensus sites in SIN proteins, or have any SIN components been identified at Pmk1 targets? Some additional consideration of the underlying mechanism would strengthen the framework for future studies.

Except for the Clp1p phosphatase, none of the SIN components has been identified as Pmk1p substrates. Pmk1p phosphorylates Clp1p influencing its redistribution (from the nucleolus to the nucleoplasm) upon genotoxic stress (Broadus, 2012 #5805). However, our data indicates that Clp1 is not required to delay cytokinesis in the presence of BP. Whether Pmk1p phosphorylates any of the SIN components has to be tested in the future. Accordingly, we have found several MAP kinase docking motifs (D-motifs) in Sid1p and Sid2p kinases. These motifs (R/K)xxxx#x# (where # is a hydrophobic residue) represent potential sites of interaction with the ERK1/2 and p38 subfamilies of MAP kinases (Pmk1p orthologues).

4) Some additional analysis of double mutants would strengthen the conclusions regarding the pathway that signals to Pmk1. For example in Figure 4, I expected to see the phenotype for pmk1∆ rgf1∆ double mutants, with the prediction that they do not exhibit an additive phenotype on the checkpoint.

We have included results that show *rgf1*∆ and *pmk1*∆ mutations do not have any additive effects in the BP response (Figure 4—figure supplement 4).

Moreover, an *rgf1*Δ *pmk1*Δ double mutant exhibit no synergistic effects in the BP response (Figure 4—figure supplement 4), suggesting that the Rgf1pbranch activating Pmk1p (Figure 4B) channels an efficient cytokinesis delay that protects cells from premature abscission under BP stress.

5) The authors' experiments build a signaling pathway that leads to activating phosphorylation of Pmk1, but the pathway can be equally impacted by modulating the phosphatase Pmp1, which removes the activating phosphorylation on Pmk1. It does not seem essential for the authors to test the identity or regulation of CIP phosphatases, but they should discuss potential regulatory mechanisms in the text. It seems possible that BP or other stresses might reduce Pmp1 activity, which would facilitate increased activating phosphorylation of Pmk1 by lowering the inhibitory threshold.

We agree, we should test whether elimination of Pmp1p will suppress the cytokinesis delay in *rgf1*Δ and *pck2*Δ cells or in the *cwg1-1* mutant cells. In the current manuscript, we have included a new paragraph collecting this possibility.

Discussion:

“A quick response is required when the cell’s genome has already split and the cell becomes ready to separate its cytoplasm. Cell wall sensors/transducers detect the perturbation in the septum as it starts to form and activate Pmk1p to block cytokinesis progression under the situation of a deficient septum. The pathway can be equally impacted by modulating the phosphatase Pmp1p, which removes the activating phosphorylation on Pmk1p (Sugiura, Toda, Shuntoh, Yanagida, and Kuno, 1998).”

6) Figure 5B could be improved by adding more labels to distinguish WT versus pmk1∆ data.

OK, it is done.

7) The connection between the checkpoint pathway and the septum defect in Figure 5 remains murky. The authors suggest that the checkpoint induces "the quality control machinery for septum assembly and/or disassembly." I am not aware of evidence that these cells build "a defective septum that was difficult to digest by the glucanases." This speculation is consistent with their data but should be phrased much more hypothetically.

We build on previous results. It was known that under conditions of nutrient limitation, hypertonic stress or caffeine, *pmk1*∆ cells grow as short-branched filaments in which the cell walls and septa are thickened (Zaitsevskaya-Carter, 1997 #982). Apparently, these cells had almost finished cytokinesis, but had not completely lysed the external wall (Zaitsevskaya-Carter) (Zaitsevskaya-Carter, 1997 #982).

We have reorganize this part a little bit. We have eliminated the sentence and we have included a new paragraph in the Discussion:

“In the case that the checkpoint is overridden, some septa could be poorly built up. Under conditions of nutrient limitation, hypertonic stress or caffeine, *pmk1*∆ cells grow as short-branched filaments in which the cell walls and septa are thickened (Zaitsevskaya-Carter and Cooper, 1997). At the same time, Pmk1p might induce the expression of genes that function to replace the damaged wall and restart cytokinesis even under stress conditions (D. Chen et al., 2003; Takada et al., 2007).”

8) From experiments combining myp2∆ with BP, the authors conclude that "there could be a synergy between the cell wall stress and Myp2 motor action." I recommend that the authors reword this interpretation to explain that Myp2 and BP act independently on the timing of cytokinesis, as shown by the additive defect of myp2∆ and BP.

OK, it has been done.

The sentence “This suggested there could be a synergy between the cell wall stress and Myp2 motor action” has been changed to “This suggested that the cell wall stress and Myp2 motor action act independently on the timing of cytokinesis”.

9) The authors conclude that the CAR is more stable upon BP treatment based on low-dose LatA experiments. A more direct test of CAR stability would be the timing of disassembly upon high-dose LatA treatment. The two tests (high and low dose) are complementary and would provide a stronger result in combination. The low-dose experiment sets up a competition between CAR and other actin structures, as opposed to directly testing how quickly the CAR actin turns over.

For technical reasons we have not been able to test high doses of LatA on the stability of the ring. The proportion of cells with medial F-actin structures has to be analyzed within minutes after addition of LatA (200µm), and we need to tune up a fixation protocol that preserves the Life-act-GFP fluorescence.

10) I do not find the band shift experiments in Figure 6E convincing. One possibility is to run all of the 120 timepoint samples next to each other. As presented, the subtle band shift leaves comparison of different samples quite challenging. This figure also needs a phosphatase-treated control to establish the amount of band shift in the authors' system.

We have tested different acry/bis gel percentage and running conditions trying to get a better band shift. Moreover, we have repeated the experiment at least 4 times with similar results (see Author response image 7). In the presence of BP, the samples of 90 and 120 minutes always show slightly higher mobility indicative of a sort of “hipophosphorylated form”. However, in wild type (w/o BP) and *pmk1*Δ (+BP) cells the 120-minute sample is always running a little bit slower than the 90 minutes sample. It is probable that 2h after release most cells have completed septation and carry Cdc15p in a more phosphorylated form. The two replicas are shown in the supplemental material (Figure 6—figure supplement 3).

**Author response image 7. respfig7:** 

11) The authors use fixed timepoint microscopy in Figure 7A to conclude that "ring disassembly dynamics was similar in the presence of absence of BP" for sid2-250 mutants. It seems important to perform timelapse imaging and to measure disassembly dyamics for individual rings, if the authors want to draw this conclusion.

We have rephrased the sentence to discard the idea that the ring disassembly dynamics was similar in the presence or absence of BP for *sid2-250* mutants. This is included in the revised version:

“To address this question, *sid2-250* cells expressing Imp2p-GFP and Sid4p-GFP were shifted to a restrictive temperature and treated with BP (10 µg/ml) 30 minutes prior to being sampled at 2 and 3 hours. Interestingly, the number of rudimentary rings was similar in the presence or absence of BP (Figure 7A), suggesting that the SIN might be involved in CAR stabilization in the presence of cell wall stress.”

12) Figure 7B provides a very convincing quantification of Cdc7 dynamic localization at SPBs. I would recommend that the authors present a similar graph for the data in Figure 7—figure supplement 1.

We initially try to measure Cdc7p-GFP intensity in Figure 7—figure supplement 1, but, since it shares the same fluorescence as Imp2-GFP, there was too much interference between Imp2p-GFP cytoplasmic/scattered fluorescence and that of Cdc7p. Then, we decided to improve the conditions by removing the ring marker and adding a second SPB marker in another fluorescent channel, which made the quantifications more reliable in Figure 7B. We decided to keep Figure 7—figure supplement 1, as an extension to the quantifications in Figure 7B, where the CAR and septum can be seen in combination with Cdc7p-GFP.

Reviewer #3:The Rho-GTPase pathway has been implicated in cytokinesis with roles in regulation of the contractile actomyosin ring (CAR), cell-wall synthesis (in budding and fission yeasts), and other signaling pathways. In this paper, the authors showed that the Rho1 GEF Rfg1 localizes in the region following the contractile ring during cell division, in a CAR-dependent manner. Interestingly, the authors found that treatment of cells with the BP dye that specifically binds to linear 1,3-β-glucan delays cytokinesis in WT, but not in rfg1∆ cells. The delay was largely in the timing of onset of CAR constriction and concomitant ingression of plasma membrane, rather than the rate of constriction/ingression. Based on these results, the authors hypothesize that Rfg1 is involved in a mechanism that senses cell-wall perturbation and delays the onset of cytokinesis. Because Rfg1 has previously been implicated in the cell-wall integrity pathway (CIP), the authors tested the requirement of known CIP components for BP-responsive cytokinesis delay, and identified Rho1p, Pck2p, and Pmk1p as downstream factors of Rfg1p in this response. pmk1∆ cells show significant increase in septation index upon BP treatment, which the authors conclude to be a consequence of failure to delay cytokinesis onset. The delay in the onset of cytokinesis correlated with prolonged cytoplasmic retention of the Clp1 phosphatase and dephosphorylation of its substrate Cdc15. Finally, the delay in cytokinesis onset requires normal SIN activity, implicating the pathway in this process.Contribution of cell wall to cytokinesis has been clearly demonstrated in both fission yeast and budding yeast, yet its regulation and coordination with the CAR pathway has not been fully established. This paper provides a new insight into the ability of the cells of *S. pombe* in controlling CAR constriction, septum synthesis, and membrane ingression in response to perturbation to cell-wall synthesis. Most of the experiments are performed carefully and data are clearly presented, which are of interest to the readers in the field.1) While the manuscript clearly demonstrates the cell's ability to respond to cell-wall perturbation that involves delayed onset of CAR constriction and septum synthesis, the biological significance of this observation is unclear. How did *S. pombe* evolve to have this adaptation pathway, and what is the consequence of a defect in it?

Fission yeast cells probably evolved from filamentous fungi (Sipiczki, 1995 #1378). Since the absence of Pmk1p leads to a hyphal-like growth pattern under certain stress conditions, it is possible that the CIP contributed to the emergence of proliferation through single cells. In addition, this regulatory pathway can be modulated to fine-tune growth modes depending on environmental conditions. Under conditions of starvation, it may be advantageous to grow in a hyphal mode organism to efficiently capture nutrients.

Although the authors conclude that the observed separation defect is a consequence of the failure to delay cytokinesis onset, the causality between the two phenotypes does not seem to be fully established. The separation defect could be a separate consequence of the defective CIP under cell-wall stress, such as through some defects in the secretory pathway, that is independent of the delayed cytokinesis onset.

It is known for a long time that *pmk1*∆ cells grow as short-branched filaments under certain stress conditions (nutrient, high temperature, osmotic stress …). Apparently, these cells had almost finished cytokinesis, but had not completely lysed the external wall, suggesting defects in cell wall remodeling. (Zaitsevskaya-Carter, 1997 #982). However, to date there are not effectors of this pathway with a known function in cytokinesis.

In this paper we show for the first time that a cell wall stressor (BP induces Pmk1p within minutes (Barba, 2008 #5646;Madrid, 2006 #5648)) triggers a checkpoint-like response during cytokinesis. The response is overridden in the absence of the kinase that channels the damage. Although we found a good correlation between cell wall stress and CIP induction, we cannot demonstrate that the separation defects observed in CIP mutants are exclusively dependent on their inability to activate the checkpoint.

Thus, we have included the changes shown below.

“Thus, the chained-cell phenotype was probably due to the mutant´s inability to halt cytokinesis and induce the quality control machinery for septum assembly and/or disassembly.”

Discussion:

“In the case that the checkpoint is overridden, some septa could be poorly built up. In that sense it is known that conditions of nutrient limitation, hypertonic stress or caffeine, prompt *pmk1*∆ cells to grow as short-branched filaments in which the cell walls and septa are thickened (Zaitsevskaya-Carter and Cooper, 1997). At the same time, Pmk1p might induce the expression of genes that function to replace the damaged wall and restart cytokinesis even under stress conditions (D. Chen et al., 2003; Takada et al., 2007).”

Do any of the other CIP pathway mutants show separation defects, and if they do, is there a good correlation between the timing of Imp2 constriction (Figure 4) and separation defects?

Null mutants of Mkh1p, Skh1p/Pek1p and Pmk1p show separation defects when grown in nutrient-limiting conditions at high temperature or in hyperosmotic medium (Sengar, Markley, Marini, and Young, 1997; Toda et al., 1996b; Zaitsevskaya-Carter and Cooper, 1997). Null mutants of Pck2p do not multi-septate but instead cells lyse as doublets during cytokinesis. Regarding Rgf1p, 15% of the null cells lyse and 70% show monopolar growth; these cells are impaired for growth through the new end that comes from the anterior division, suggesting problems in the recognition of a faulty disassembled end (Bohnert, 2012 #5592). Moreover, mutants with late cytokinesis defects likewise frequently exhibit new end growth polarity errors (Bohnert, 2012 #5592).

Do mutations in myp2 etc. that delay the onset of constriction also rescue the separation defect in pmk1∆? Can the authors speculate as to whether the cell-separation defect causes a severe morphological or ecological disadvantage that explains the evolution of this response pathway?

In our hands, Myp2 mutations do not rescue the cell separation defect in *pmk1*∆ cells. The double mutant *pmk1*Δ *myp2*Δ shows higher percentage of multiseptated cells than any of the individual mutants *pmk1*Δ and *myp2*Δ (not shown). This result also supports that Myp2p and BP act independently on the timing of cytokinesis.

2) In many figures, rgf1∆ and pmk1∆ are used interchangeably to test the role of the proposed pathway in delaying cytokinesis onset. While this is consistent with the previously established CIP pathway, in which Rgf1 and Pmk1 are positioned at the most upstream and downstream (Figure 4B), it is possible that the two proteins function independently in separate or partially overlapping pathways leading to cytokinesis delay in BP-treated cells. Thus, the authors should at least establish that rgf1∆ and pmk1∆ mutations do not have any additive effects in the BP response, and ideally repeat some key experiments in both mutant backgrounds.

We have included results that show *rgf1*∆ and *pmk1*∆ mutations do not have any additive effects in the BP response (Figure S4E).

“Moreover, an *rgf1*Δ *pmk1*Δ double mutant exhibit no synergistic effects in the BP response (Figure 4—figure supplement 4), suggesting that the Rgf1pbranch activating Pmk1p (Figure 4B) channels an efficient cytokinesis delay that protects cells from premature abscission under BP stress.”